# Smooth muscle-specific MMP17 (MT4-MMP) regulates the intestinal stem cell niche and regeneration after damage

Mara Martín-Alonso [1✉], Sharif Iqbal[2,3], Pia M. Vornewald [1], Håvard T. Lindholm [1], Mirjam J. Damen[4], Fernando Martínez [5,6], Sigrid Hoel[1], Alberto Díez-Sánchez [1], Maarten Altelaar [4], Pekka Katajisto [2,3,7], Alicia G. Arroyo [8,9] & Menno J. Oudhoff [1✉]

Smooth muscle is an essential component of the intestine, both to maintain its structure and produce peristaltic and segmentation movements. However, very little is known about other putative roles that smooth muscle cells may have. Here, we show that smooth muscle cells may be the dominant suppliers of BMP antagonists, which are niche factors essential for intestinal stem cell maintenance. Furthermore, muscle-derived factors render epithelium reparative and fetal-like, which includes heightened YAP activity. Mechanistically, we find that the membrane-bound matrix metalloproteinase MMP17, which is exclusively expressed by smooth muscle cells, is required for intestinal epithelial repair after inflammation- or irradiation-induced injury. Furthermore, we propose that MMP17 affects intestinal epithelial reprogramming after damage indirectly by cleaving diffusible factor(s) such as the matricellular protein PERIOSTIN. Together, we identify an important signaling axis that establishes a role for smooth muscle cells as modulators of intestinal epithelial regeneration and the intestinal stem cell niche.

[1] Centre of Molecular Inflammation Research, and Department of Clinical and Molecular Medicine, Norwegian University of Science and Technology, Trondheim, Norway. [2] Institute of Biotechnology, HiLIFE, University of Helsinki, Helsinki, Finland. [3] Molecular and Integrative Bioscience Research Programme, Faculty of Biological and Environmental Sciences, University of Helsinki, Helsinki, Finland. [4] Biomolecular Mass Spectrometry and Proteomics, Bijvoet Center for Biomolecular Research and Utrecht Institute for Pharmaceutical Sciences, Utrecht University, Utrecht, Netherlands. [5] Bioinformatics Unit. Centro Nacional de Investigaciones Cardiovasculares (CNIC), Madrid, Spain. [6] Centro de Investigación Biomédica en Red de Enfermedades Cardiovasculares (CIBERCV), Madrid, Spain. [7] Department of Cell and Molecular Biology, Karolinska Institutet, Stockholm, Sweden. [8] Department of Molecular Biomedicine, Centro de Investigaciones Biológicas Margarita Salas (CIB-CSIC), Madrid, Spain. [9] Vascular Pathophysiology Area, Centro Nacional de Investigaciones Cardiovasculares (CNIC), Madrid, Spain. ✉email: mara.m.alonso@ntnu.no; menno.oudhoff@ntnu.no

The intestinal epithelium consists of a single layer of cells that is important for the uptake of nutrients as well as for providing a barrier to protect from pathogens. During homeostasis, intestinal epithelial cells are all derived from LGR5+ intestinal stem cells (ISCs) that reside at the bottom of crypts[1]. Upon injury, however, or after depletion of LGR5+ cells, LGR5- cells can rapidly regain LGR5 expression and thus dedifferentiate to repopulate the crypt bottoms[2,3]. In addition, within this dedifferentiation process, an epithelial reparative state exists, which is fetal-like, and depends on reprogramming by YAP[4,5], and is further characterized by markers such as SCA-1 and HOPX[4,6,7].

Adult intestinal epithelial (stem cell) maintenance relies on a variety of niche factors such as WNTs, R-spondins (RSPOs), Bone morphogenic proteins (BMPs), and prostaglandins, all of which are expressed by mesenchymal cell subtypes[8–13]. These mesenchymal cells reside in the intestinal lamina propria between the epithelium and the muscularis mucosae. Here we find several cell types characterized by specific marker expression and ultrastructural features in electron microscopy such as different types of telocytes (FOXL1[14,15],PDGFRA high), trophocytes (PDGFRA low[8]), fibroblasts, myofibroblasts (SMA+, Desmin−), and mesenchymal stromal/stem cells[16,17]. However, intestinal epithelial homeostasis does not solely rely on soluble niche factors. The mechanical or extracellular matrix (ECM) niche is an additional defining factor, for example, by modulating the mechanosensory HIPPO/YAP pathway[4,18]. In addition, growth factors can interact with, or be embedded within the ECM to modulate their activity. Smooth muscle cells reside in the muscularis mucosae, as a layer underneath the lamina propria, as well as outside of the mucosa in the muscularis propria in two thick layers, circular and longitudinal, and can be distinguished by being SMA and Desmin positive[16,17,19]. Smooth muscle cells are one of the most prevalent non-epithelial cell types throughout the intestine, yet, their role in providing niche factors or affecting the ECM niche is largely unknown. Nevertheless, smooth-muscle-specific deletion of tumor suppressor genes can result in defective epithelial growth[20]. Furthermore, it was shown that cells that originated from the smooth-muscle can migrate into the mucosa to aid after injury[21]. Nevertheless, it is still largely undefined what role adult smooth muscle has in ISC maintenance or during repair after injury.

Matrix metalloproteinases (MMPs) are fundamental ECM regulators, both by modifying ECM components directly and by cleaving growth factors to control their ability to bind the ECM and to the cell[22–24]. MMPs can play various roles in the injured intestine and many MMPs are upregulated in inflammatory bowel disease, likely by the increase in immune cell populations such as neutrophils with high proteolytic activity or in endothelial cells[25,26]. Thus, inhibition of MMP activity may be an attractive therapeutic target for treating inflammatory bowel disease[26,27]. Although the role of certain MMPs such as MMP2, MMP7, MMP9, and MT1-MMP are relatively well-studied, the role of other MMPs in intestinal biology is still largely unknown. Here, we show that MMP17, a membrane-bound MMP which is specifically expressed in smooth muscle, is important to maintain optimal ISC stemness during homeostasis, and preserve the regenerative capacity of intestinal epithelium.

## Results

### Intestinal smooth muscle is a rich source of BMP antagonists.
Based on recent indications that fetal intestinal muscle is a provider of niche factors[28], we first isolated the smooth muscle tissue from the muscularis propria (circular and longitudinal muscle) and part of the muscularis mucosae of adult mouse colon and performed RNA-seq comparing it to isolated colonic crypts

(Figs. 1a, b, S1a, S1b). We found little evidence that intestinal smooth muscle expressed niche factors such as WNTs and RSPOs, or growth factors such as epidermal growth factor (EGF), however, we did find that smooth muscle expresses high levels of factors associated with BMP signaling including *Grem1, Grem2*, and *Chrdnl1* (Figs. 1b, S1b). Fluorescent in situ hybridization (FISH) confirmed high levels of these factors in a muscle-specific manner, in particular, we found enrichment of these factors in the muscularis mucosa that resides in close proximity to the bottom of epithelial crypts (Fig. 1c).

### Intestinal smooth muscle provides niche factors that render organoid growth independent of NOGGIN.
Intestinal organoids (including small-intestinal organoids derived from the duodenum (SI organoids) and colon-derived organoids (colonoids)) are self-organizing in vitro epithelial structures that are widely used to model in vivo processes[29–31]. To test for a functional role of smooth muscle cells in providing niche factors, we exposed SI organoids to supernatant from muscle explants (muscle-SN) that completely lacked the mucosa (Fig. S1a). We found that organoids exposed to muscle-SN grew 2–3 times larger than in standard conditions and had a predominantly spheroid morphology (Fig. 1d–g). SI organoids rely on the supplementation of 3 factors epithelium itself does not express sufficiently: EGF, NOGGIN, and RSPO1 (ENR). Thus, we next tested whether muscle-SN could replace these factors. We found that muscle-SN was unable to replace RSPO1, as SI organoids failed to grow in RSPO1-deficient medium irrespective of muscle-SN supplementation (Fig. 1d, e). In contrast, we found that muscle-SN was able to substitute for the BMP antagonist NOGGIN (Fig. 1d, e). This is in support of our RNA-seq data in which we observed low *Rspo1/2/3* levels, but ample expression of BMP antagonists such as *Grem1* in smooth muscle tissue (Figs. 1b, c, S1b). ENR medium supports self-renewal and differentiation of SI organoids that acquire a typical structure including proliferative budding crypts and non-proliferative villus regions (Fig. 1f, g). In contrast, muscle-SN exposed SI organoids lacked crypts and consisted of mainly proliferative cells that were equally distributed along the spheroid (Fig. 1f, g). Our findings are not unique for colonic smooth muscle explants as we obtained similar results with muscle-SN derived from the small intestine (Figs. S2a, S2b).

### MMP17 is a smooth muscle-specific MMP that is enriched in the muscularis mucosa.
We find that intestinal smooth muscle expresses and secretes soluble niche factors that can mediate epithelial organoid growth (Fig. 1). Next, we wished to determine whether smooth muscle may also affect the ECM. The ECM acts on ISCs by providing a 'mechanical' niche as well as by being a reservoir of ECM-bound niche factors[23,32,33]. MMPs are known as important modulators of the ECM[22,34]. By directly comparing epithelial crypt and smooth muscle we found several MMPs to be specifically expressed in smooth muscle tissue (Fig. 2a), including *Mmp17* (also known as MT4-MMP), a previously identified regulator of muscle growth factors and its surrounding ECM[24]. Therefore, we analyzed *Mmp17* expression in intestinal tissue by using the KO/KI mouse strain *Mmp17*[LacZ/LacZ] (referred to as KO mice from hereon)[35]. We used LacZ staining (Fig. S3a) or a specific anti β-gal antibody to detect *Mmp17* promoter activity in the intestine from *Mmp17*[LacZ/+] mice. As shown in Fig. 2b, MMP17+ cells are exclusively smooth muscle cells, SMA+ and DESMIN+ (Fig. 2b, Fig. S3b). Of these, we observed more smooth muscle MMP17+ cells in the muscularis mucosae compared to the muscularis propria (longitudinal and circular muscle layers) (Fig. 2c). The expression pattern is reminiscent of the expression of BMP antagonists (Fig. 1c), and indeed, using FISH,

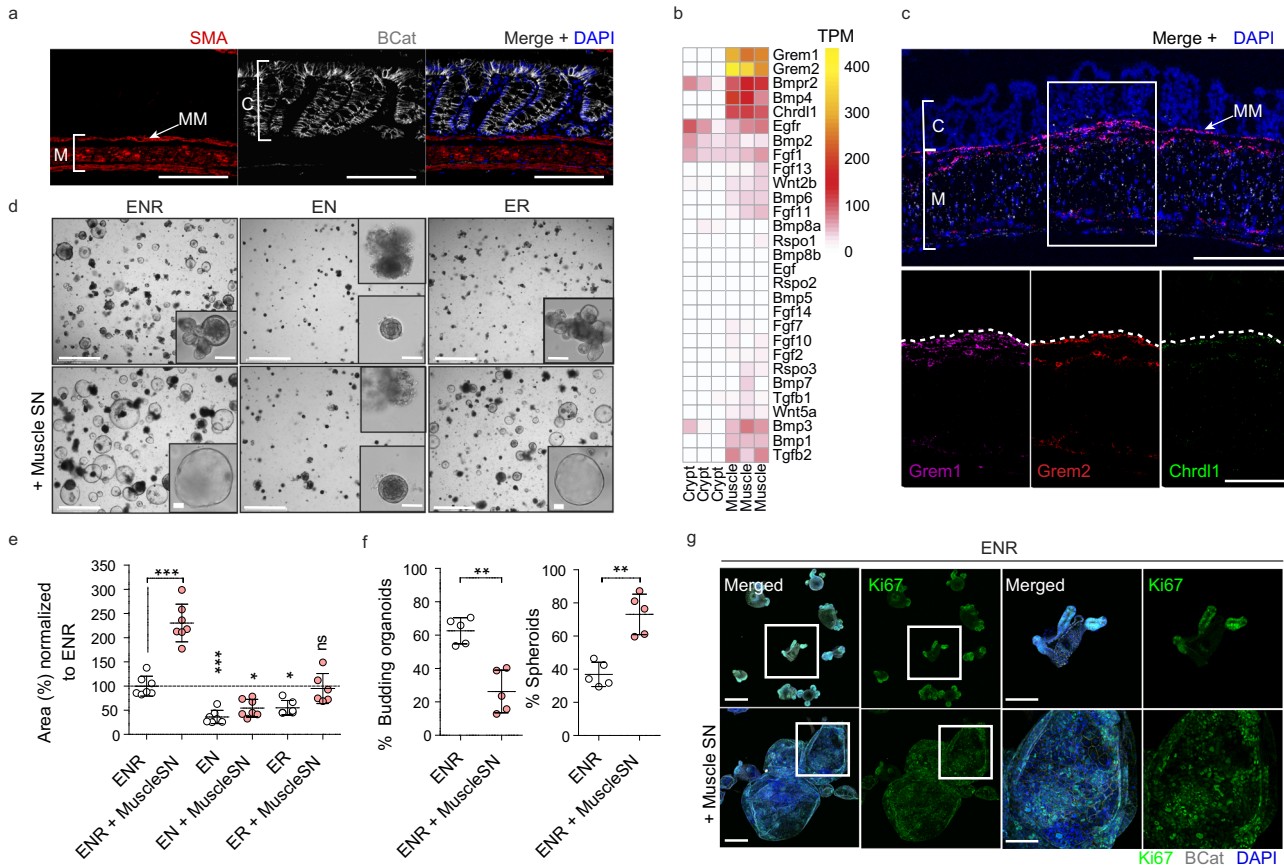

**Fig. 1 Intestinal Smooth muscle expresses BMP antagonists that control epithelial behavior. a** Representative confocal maximum intensity projection of a transverse colon cut showing specific staining for muscle (M), SMA (Red), and crypts (C), βCatenin (βcat, Gray). Clear distinction among muscularis mucosa (MM) close to the crypts and the muscularis propria (circular and longitudinal smooth muscle) can be observed. Scale bar 100 µm. $n = 4$ mice. **b** Heatmap depicting gene expression levels in TPM (Transcripts per million) in crypts vs muscle intestinal samples. $n = 3$ biological replicates. **c** Representative confocal maximum intensity projections of fluorescence-coupled RNAscope showing BMP antagonists *Grem1* in magenta, *Grem2* in red, and *Chrdl1* in green, expressed in smooth muscle cells. Scale bar 100 µm; 50 µm in the magnified image. $n = 2$ mice. **d** Representative bright-field pictures showing SI organoids morphological changes when exposed to muscle-derived factors (Day4). Control media ENR (EGF, NOGGIN, RSPO1), EN (no RSPO1) or ER (no NOGGIN) were used to assess SI organoids reliance on external growth factors. Scale bar 1250 µm; 100 µm in magnified image. **e** Graph shows SI organoids area in response to different media. Depicted average values normalized to ENR condition. $n = 6$ (for ER and ER + MuscleSN) to 7 (the rest) independently performed experiments. Each dot corresponds to the average of 2–3 wells/experiment. **f** Graphs represent the percentage of budding organoids or spheroids in response to ENR or ENR+ muscle supernatant (Muscle-SN). $n = 5$ independently performed experiments with 2–3 wells/ experiment. **g** Representative maximum projection confocal images showing proliferative active cells (Ki67) in green and cell shape (βcat, gray) staining. Scale 200 µm, 100 µm in magnified image. $n = 3$ independent experiments. Numerical data are means ± SD. Data were analyzed by one-way ANOVA ($F = 57.06$, $p < 0.0001$) followed by Bonferroni's multiple comparison test (ENR vs ENR + Muscle SN ***$p < 0.001$; ENR vs EN ***$p < 0.001$; ENR vs EN + Muscle SN *$p < 0.05$; ENR vs ER *$p < 0.05$; ENR vs ER + Muscle SN, ns, not significant.) (**e**) or by Mann–Whitney–test, two-tailed (**$p = 0.0079$ in both) (**f**). Source data are provided as a Source Data file.

we find that *Mmp17* has an overlapping expression pattern with these BMP antagonists including enrichment in the muscularis mucosa (Fig. 2d).

**Muscle-specific MMP17 controls BMP signaling in crypts.** Since MMP17 is enriched in the muscularis mucosa smooth muscle cells where its expression correlates with BMP antagonists, we asked about the possible impact of MMP17 loss in crypts. To unbiasedly gain mechanistic insight, we performed RNA-seq comparing WT and KO colonic smooth muscle and crypts. *Mmp17* is expressed in smooth muscle cells but not in the epithelium (Fig. 2), and we were surprised to find that only 42 genes were dysregulated in the KO smooth muscle whereas 191 genes were dysregulated in KO crypts compared to their WT counterparts (Fig. 3a). In support, principal component analysis (PCA) was unable to distinguish WT from KO smooth muscle, whereas KO crypts had a different distribution

compared to WT crypts (Fig. 3b). Furthermore, *Mmp17* absence in intestinal smooth muscle cells did not result in any structural alteration at the smooth muscle level (Fig. S4a). Upon closer examination, using the online gene enrichment tool Enrichr[36,37], we found SMAD4 as the top TF associated with upregulated genes in KO crypts (Figs. 3c, S4c). To test whether increased SMAD4 target genes in KO crypts were a direct result of SMAD4 protein levels, we performed immunostaining and western blot for SMAD4. Corroborating our unbiased transcriptome analysis, we found heightened nuclear localization of SMAD4 in the bottom of KO crypts compared to WT crypts, and increased levels of total SMAD4 in KO mucosa (Fig. 3d, e). The nuclear translocation or overall accumulation of SMAD4 is the result of cellular activation by TGFβ family members, including BMPs that specifically induce SMAD1/5/9 phosphorylation[38]. Indeed, we found pSMAD1/5/9 to be particularly enriched in the bottom of KO crypts compared to WT

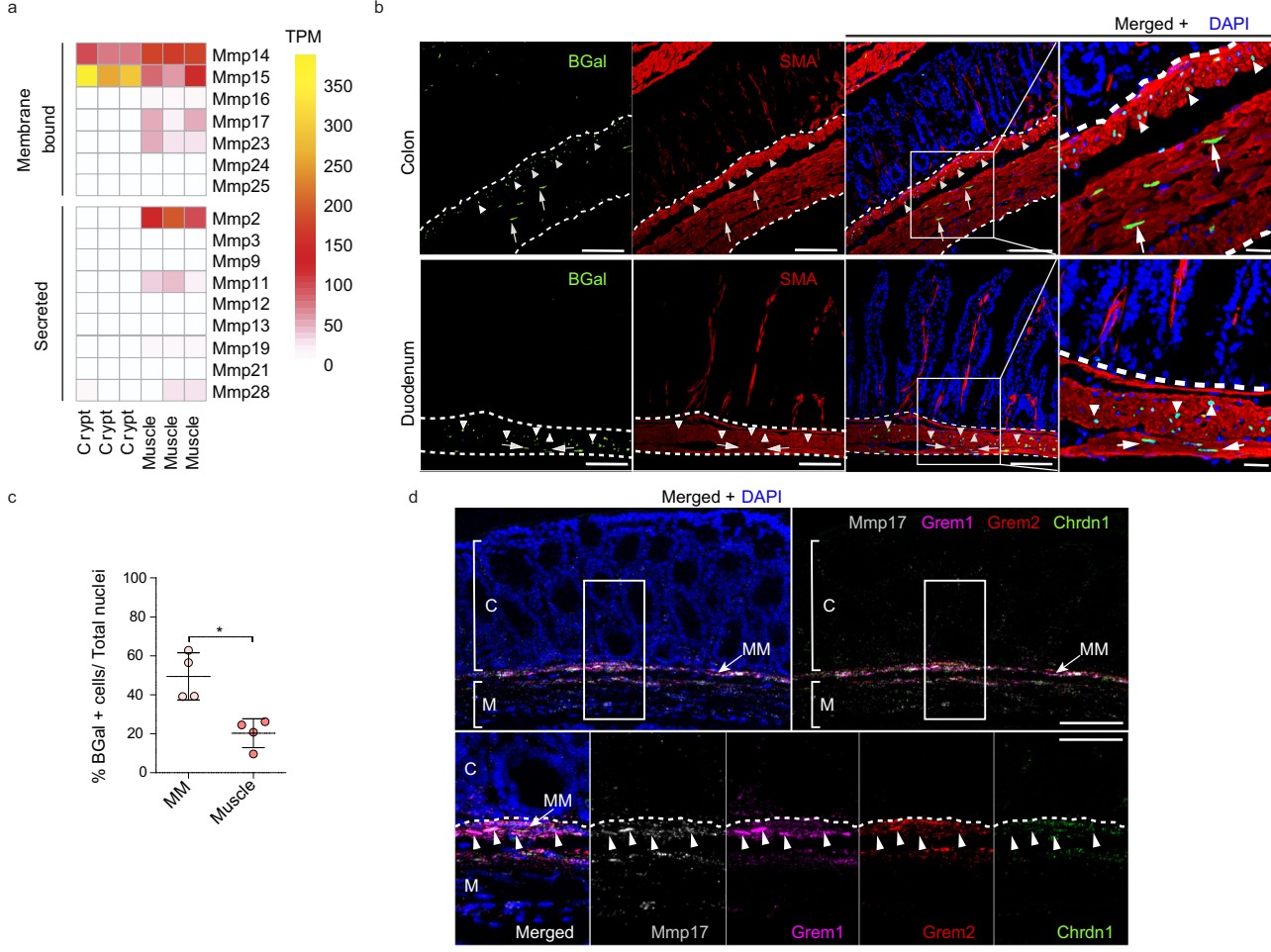

**Fig. 2 Muscle-specific matrix metalloproteinase Mmp17 is enriched in the muscularis mucosa together with BMP antagonists expressing cells.**
**a** Heatmap shows TPM levels of Matrix Metalloproteinases (MMPs) in crypts vs muscle. $n = 3$ biological replicates. **b** Representative confocal microscopy images showing active Mmp17 promoter (βGal staining, green) in positive SMA (red) muscle (white dashed line) stained in transverse intestinal sections of *Mmp17±* mice. Scale bar 100 μm; 20 μm in the magnified image. n = 3 animals. **c**. Graph shows quantification of the % number of βGal positive cells in muscularis mucosa (MM) vs circular and longitudinal muscle (Muscle) in colon samples. $n = 3$ biological replicates. Four images quantified.
**d** Representative confocal maximal projections of fluorescence-coupled RNAscope showing *Mmp17* (gray), and BMP antagonists (*Grem1*, magenta, *Grem2*, red and *Chrdl1*, green) co-expression in muscular cells (arrowheads). C, crypts, M, Muscle and MM, muscularis mucosa. Scale bar 100 μm; 50 μm in the magnified image. $n = 2$ independent experiments with 1–2 samples/genotype. Numerical data in (**c**) are means ± SD and was analyzed by Mann–Whitney test (two-sided, *$p = 0.0286$). Source data are provided as a Source Data file.

crypts (Fig. 3f), which suggests altered BMP signaling in KO intestinal epithelium.

**MMP17 regulates the ECM-ISC niche necessary for epithelial de novo crypt formation.** SMAD signaling is essential to maintain ISCs, which is exemplified by the requirement of the BMP antagonist NOGGIN in intestinal organoid maintenance[29]. To test if MMP17 affects ISCs, we quantified levels of the ISC markers Lgr5 and Olfm4 by ISH. We found that KO intestinal tissue had lower Lgr5 and Olfm4 levels compared to WT tissue in both the small and large intestines (Fig. 4a, b). In addition, decreased ISC marker gene expression was echoed by reduced colonoid and SI organoid formation efficiency comparing KO with WT crypts (Figs. 4c and S5, respectively). Importantly, colonoids splitting, or culturing with excessive niche factors (WNR medium), resulted in equal organoid efficiency between WT and KO cultures (Fig. 4d). These data indicate that the reduced capacity of KO crypts to form organoids relied on the in vivo niche rather than a consequence of an intrinsic epithelial defect. To formally test whether MMP17 controls the 'ECM niche', we utilized a recently

developed regeneration assay[39] in which small intestinal epithelium was cultured on decellularized small intestinal extracellular matrix (iECM). Unlike Matrigel-based closed format organoids, here, organoids engraft on the iECM by first generating an epithelial monolayer[39]. After initial attachment in a monolayer, de novo crypt formation occurs on former crypt pits, and thus is surrounded by the tissue native ECM. Since the reparative growth of the epithelium does not rely on Matrigel during the culture on iECM—this system was used to investigate the role of the extracellular niche on ISC function ex vivo[39]. When we cultured wild-type epithelium on WT and KO iECM, we observed a robust reduction of the de novo crypt formation on KO iECM (Fig. 4e–g). This suggests a poor regeneration supporting capacity of KO iECM.

**MMP17 is required for intestinal repair after inflammation or radiation-induced injury.** De novo crypt formation is important during epithelial repair which is needed upon injury, for example after the damage inflicted by inflammation or due to irradiation. To test if smooth muscle, and in particular MMP17, could play a

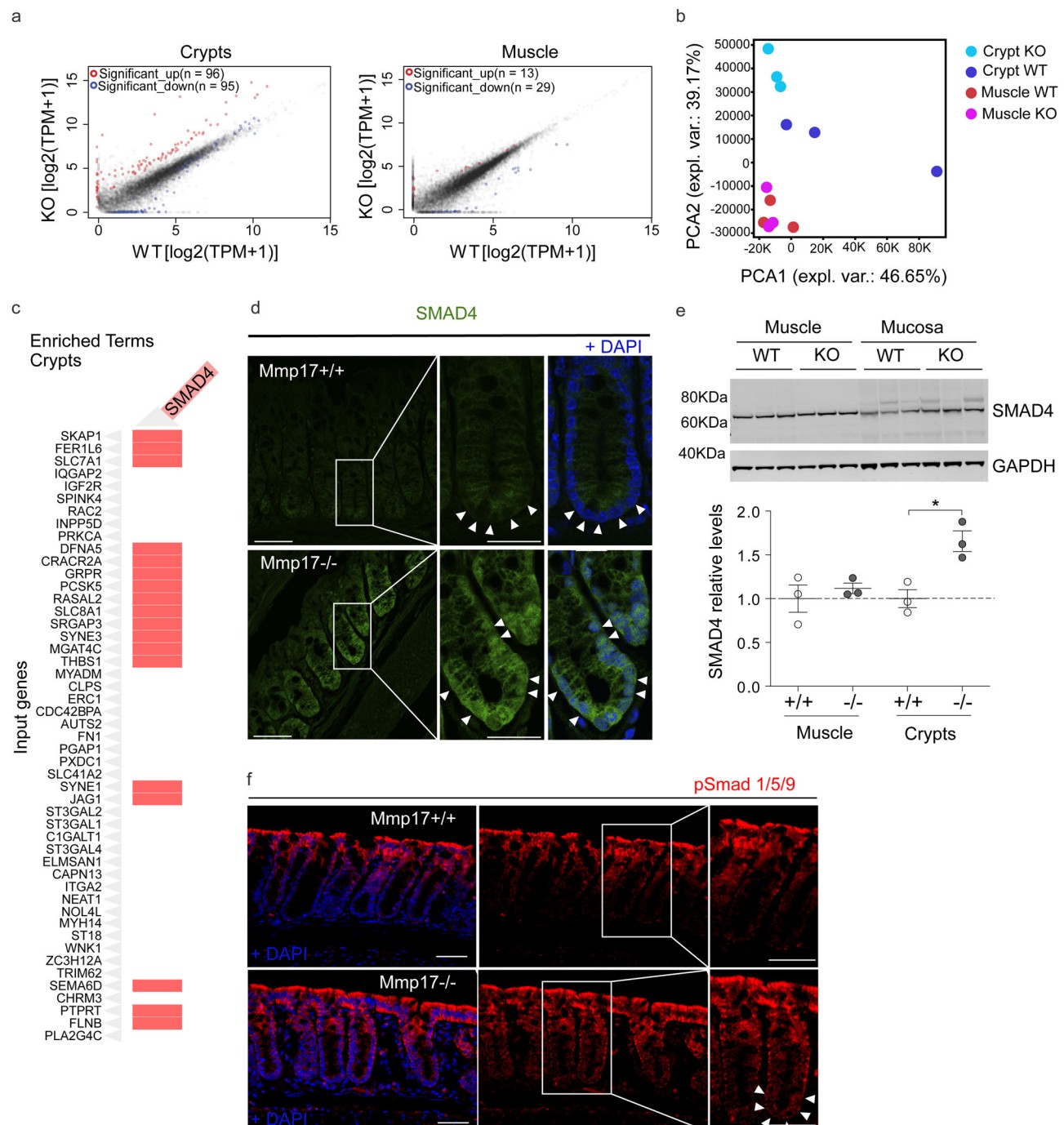

**Fig. 3 Muscle Mmp17 regulates crypts BMP signaling. a**, **b** RNA seq assay comparing WT and *Mmp17* KO crypts, we showed that loss of MMP17 in the muscle strongly affects epithelial gene expression to a greater extent than muscle gene expression. $n = 3$ biological replicates. **c** Graph shows the presence of genes related to SMAD4 signaling among the upregulated genes in WT vs KO crypts. **d**, **e** We observed higher SMAD4 signaling that was further confirmed by immunofluorescence in (**d**) (arrowheads indicate nuclear location in *Mmp17* KO) and western blot (**e**). $n = 3$ mice/genotype. Scale bar 50 μm, 25 μm in magnified views to the right. **f** Representative confocal images showing pSmad 1/5/9 staining in intestinal crypts of WT vs KO mice. Arrowheads highlight crypt base pSmad 1/5/9 staining. Scale 50 μm. $n = 3$ independent experiments with 2–4 samples/genotype. Numerical data in (**e**) are means ± SD and were tested by one-way ANOVA ($F = 7.43$, $p = 0.0106$) followed by Tukey's test (WT vs KO Crypts *$p < 0.05$). Source data are provided as a Source Data file.

role in intestinal injury responses, we used dextran sulfate sodium (DSS) to induce experimental colitis and compare WT to KO littermates (Fig. 5a). On day 5, we found that both WT and KO mice had indistinguishable features of disease indicating that damage was induced equally (Fig. S6a–e). Of note, during DSS, MMP17 (BGal) expression remained limited to smooth muscle

cells as CD45$^+$ infiltrating cells were BGal negative (Fig. S6f). After 5 days, DSS was replaced by regular drinking water to allow for intestinal repair, which is initiated rapidly and requires reprogramming of the intestinal epithelium[4,9]. Two days after DSS, KO mice had shorter colons and sustained hemorrhage, including an increased presence of blood in stool compared to

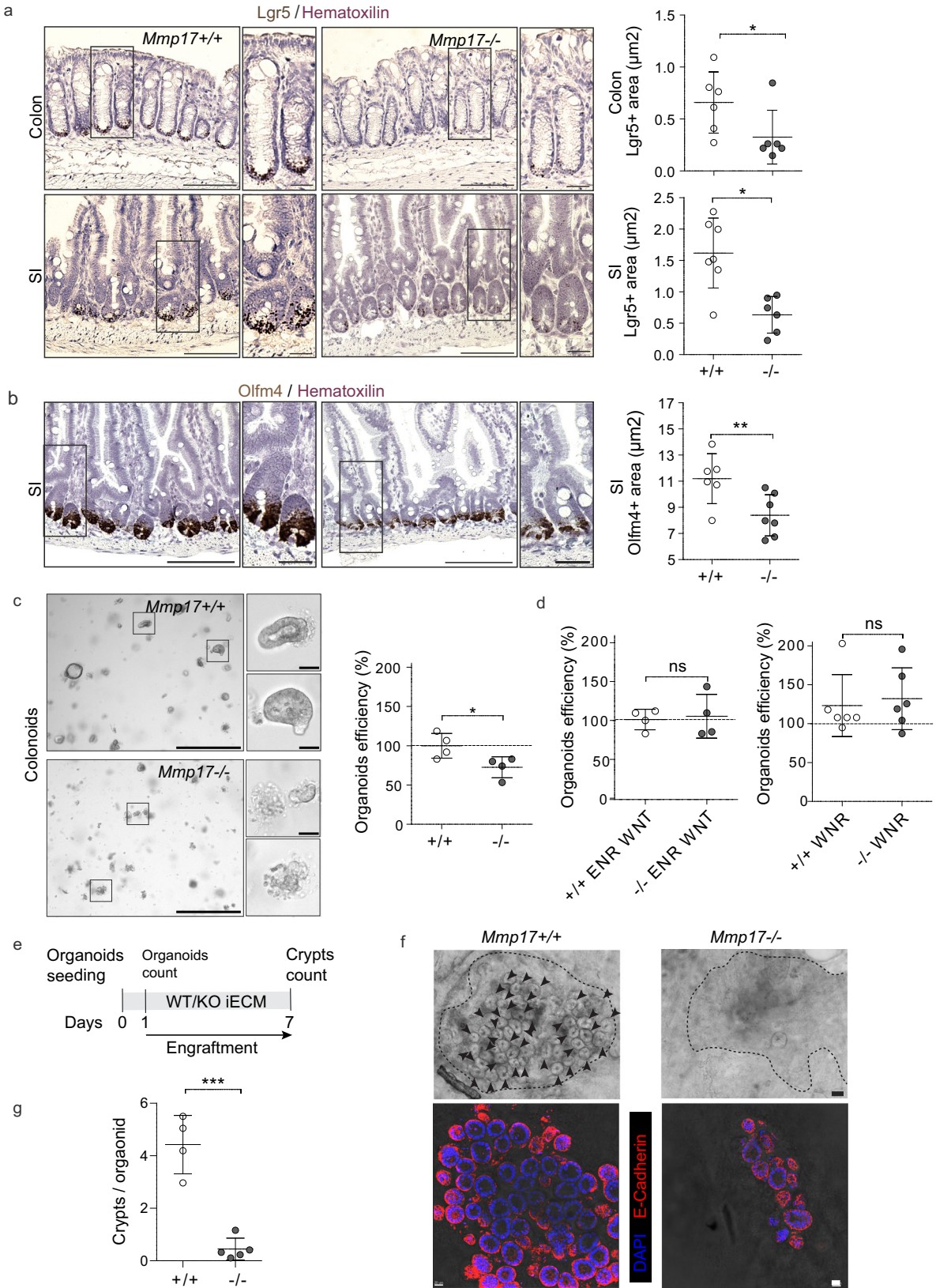

WT mice (Fig. 5b, c). Furthermore, other disease features were also exacerbated in KO mice compared to WT mice. We found that KO mice had a higher injury score than WT mice, as was determined using a genotype-blind injury classification based on H&E images (Fig. 5d, e). H&E analysis further revealed a larger area of ulcerated mucosa and lower presence of epithelial crypts in KO mice compared to WT mice (Fig. 5e, f). In addition, we found a significant reduction in the proliferative epithelium in KO mice (Fig. 5g, h), suggesting that epithelial reprogramming towards a reparative state relies on MMP17. This reprogramming it is known to rely on the activation of YAP signaling, which is characterized by an induction of (nuclear) YAP[4,5]. Indeed, we

**Fig. 4 Muscular Mmp17 regulates crypt formation. a, b** *Lgr5/Olfm4* RNAscope representative images of WT vs KO intestines and their quantification. Scale bar 100 μm; 25 μm in inset. $n = 6$ mice per genotype in Colon Lgr5 and WT Olfm4 and 7 mice per genotype in the rest. **c** Representative bright-field images of colonoids from WT vs KO crypts 48 h after crypt isolation (left). Scale 650 μm; 50 μm in cropped image. Graph shows relative colonoids number as percentage normalized to WT. $n = 5$ experiments performed. One representative experiment is showed in which each dot represents a well.,. **d** Graphs represent the percentage of colonoids after the first split (left) and the percentage of colonoids derived from colon crypts after 72 h in response to the enriched medium WNR (right). $n = $ pooled 4 wells per condition in ENR WNT experiments and 6 wells per condition in WNR experiment performed in two independent experiments. **e** Schematic of crypt formation assay on iECM; SI organoids were seeded on the iECM, counted on day 1 for attachment and survival and left to engraft into the scaffold until Day7. de novo crypts per seeded organoid (as assessed at day 1) were counted at day 7 post plating. The number of de novo crypts was counted by brightfield imaging as shown in (**f**). **f** Representative brightfield (top) and fluorescent (bottom, E-Cadherin (red), DAPI (blue), images showing epithelial crypt formation in WT and KO iECM. Arrows demonstrate the de novo formed crypts. Scale bar 50 μm (brightfield) and 20 μm (fluorescent). $n = 2$ (fluorescent) to 5 (brightfield) mice per genotype. **g** Quantification of number of crypts in WT and KO iECM. $n = 4$ WT and 5 KO mice pooled from two independent experiments (based on brightfield imaging). Numerical data are means ± SD and were tested by Mann–Whitney test (two-sided, with the following p values; $p = 0.0411$ (colon) and $p = 0.0140$ (SI) in (**a**); $p = 0.0082$ in (**b**) and $p = 0.0378$ in (**c**)) and Student's unpaired t-test (two-tailed, $p < 0.0001$ in (**g**)). Source data are provided as a Source Data file.

observe YAP-high reparative epithelium in WT tissues, however, this was nearly absent in KO tissue (Fig. 5i). This suggest that KO epithelium is unable to properly make the transition from normal to a reparative state.

Next, we performed an alternative non-inflammatory injury model based on whole-body irradiation. Indeed, a single dose of ionizing radiation (10 Gy) induces equal apoptosis in WT and KO intestines (Fig. S6g). As custom in this model, WT animals regained crypt structures 3 days after irradiation, which was delayed in KO animals and was only observed by day 6 (Fig. 5j, k). A genotype-blinded evaluation of damage features in H&E images showed increased signs of damage in small intestines of KO mice compared to WT mice 3 and 6 days after irradiation (Figs. 5l and S6h). Together, these data suggest that muscle-specific MMP17 plays a role in intestinal repair processes after damage.

***Mmp17* loss results in long-term reparative epithelial defects and increased tumor risk.** We wondered whether the intestines of KO mice would eventually heal, and thus we evaluated weight gain for a prolonged time after DSS administration (Fig. 6a, b). WT mice rapidly returned to their original weight; however, KO mice never fully regained their starting weight (Fig. 6b for females, Fig. S7a for males). In support, while WT mice largely returned to homeostatic conditions at end point, KO mice retained shorter colons, experienced sustained blood in stool, still had areas of unhealed ulcers in the epithelial surface and had higher injury scores compared to WT mice (Fig. 6c–h). In addition, we detected the presence of crypt distortions named reactive atypia, and these morphological changes were predominantly found in KO mice (Figs. 6f, g, S7b). These morphological changes resemble those found in intestinal neoplastic lesions, so we next decided to evaluate the impact of *Mmp17* loss in the initiation and progression of intestinal tumors using the *Apc*[Min] mouse model. *Apc*[Min] mice develop tumors primarily in the small intestinal epithelium[40]. We found that loss of *Mmp17* predispose mice to the formation of a higher number of tumors both quantified macroscopically (Fig. 6i), and microscopically using H&E-stained sections (Fig. 6j). We did not observe differences in tumor diameter (Fig. S7c) suggesting that MMP17 mediates tumor initiation but not tumor progression. In support, WT and KO tumors were indistinguishable in terms of β-CATENIN and OLFM4 distribution (Fig. S7d). Of note, *Mmp17* expression was restricted to muscle cells also in tumor areas (Fig. S7e). In sum, our data indicate that MMP17 is required for short and long-term intestinal epithelial repair and its loss predisposes to intestinal neoplastic alterations.

**Muscle-SN induces a reparative epithelial state in organoids via activation of YAP, and is sufficient to rescue regenerative growth of WT epithelium on KO iECM.** Since we observed that smooth muscle MMP17 is essential for intestinal repair we wondered whether muscle indeed could promote epithelial reprogramming towards a reparative state. We found that muscle-SN can replace NOGGIN in the culture media, and muscle-SN treated organoids grow as large spheroids (Fig. 1). Organoid spheroid growth can either be characterized as reparative that is associated with fetal-like gene programs, such as organoids derived from SCA-1+ cells[7], or it can be the result of increased WNT signaling such as upon treatment with the GSK3 inhibitor CHIR[41]. These two different organoid spheroid states are on opposite ends when it concerns ISCs; a reparative state is characterized by a loss of LGR5+ ISCs whereas ISCs expand upon CHIR treatment[4,7]. To determine what type of spheroids are induced by muscle-SN, we performed RNA-seq comparing normal organoids to muscle-SN treated SI organoids (Fig. 7a). Using gene set enrichment analysis, we found a distinct enrichment of gene sets associated with fetal and reparative programs in muscle-SN treated SI organoids[4,7] (Fig. 7b). Intestinal epithelial repair programs are coupled with YAP signaling and, indeed, an intestinal epithelial YAP signature gene set was also enriched in muscle-SN treated SI organoids[5] (Fig. 7b). In contrast, genes associated with LGR5+ cells[42], as well as genes upregulated in SI organoids treated with CHIR[41], were downregulated in muscle-SN treated organoids (Fig. 7b, Supplementary Data 1 includes all gene sets). In support, we found that YAP was localized nuclear throughout muscle-SN induced spheroids, whereas it was cytoplasmic in the center of budding SI organoids (Fig. 7c). In addition, in a side-by-side comparison we find that muscle-SN induces larger spheroids than treatment with CHIR, and that treatment with the YAP inhibitor Verteporfin completely abolished muscle-SN driven growth (Fig. 7d). Together, we conclude that muscle-SN induces spheroid growth that is associated with reparative/fetal-like reprogramming that may be mediated by YAP.

Next, we examined whether muscle-SN or recombinant MMP17 (rMMP17) could rescue the impaired regenerative growth of epithelium on KO iECM (Fig. 4e, f). We tested the regenerative growth on WT and KO iECM by supplementing it with muscle-SN derived from WT and KO mice or rMMP17 (Fig. 7e). We found that WT, but not KO, muscle-SN increased the crypt formation on KO iECM (Fig. 7e). Similarly, rMMP17 incubation led to increased crypt formation on KO iECM (Fig. 7e). In contrast, the different treatments of WT iECM did not affect de novo crypt formation (Fig. 7e). Taken together, these findings highlight that the enzymatic activity of MMP17 is necessary for the reparative growth of intestinal epithelium.

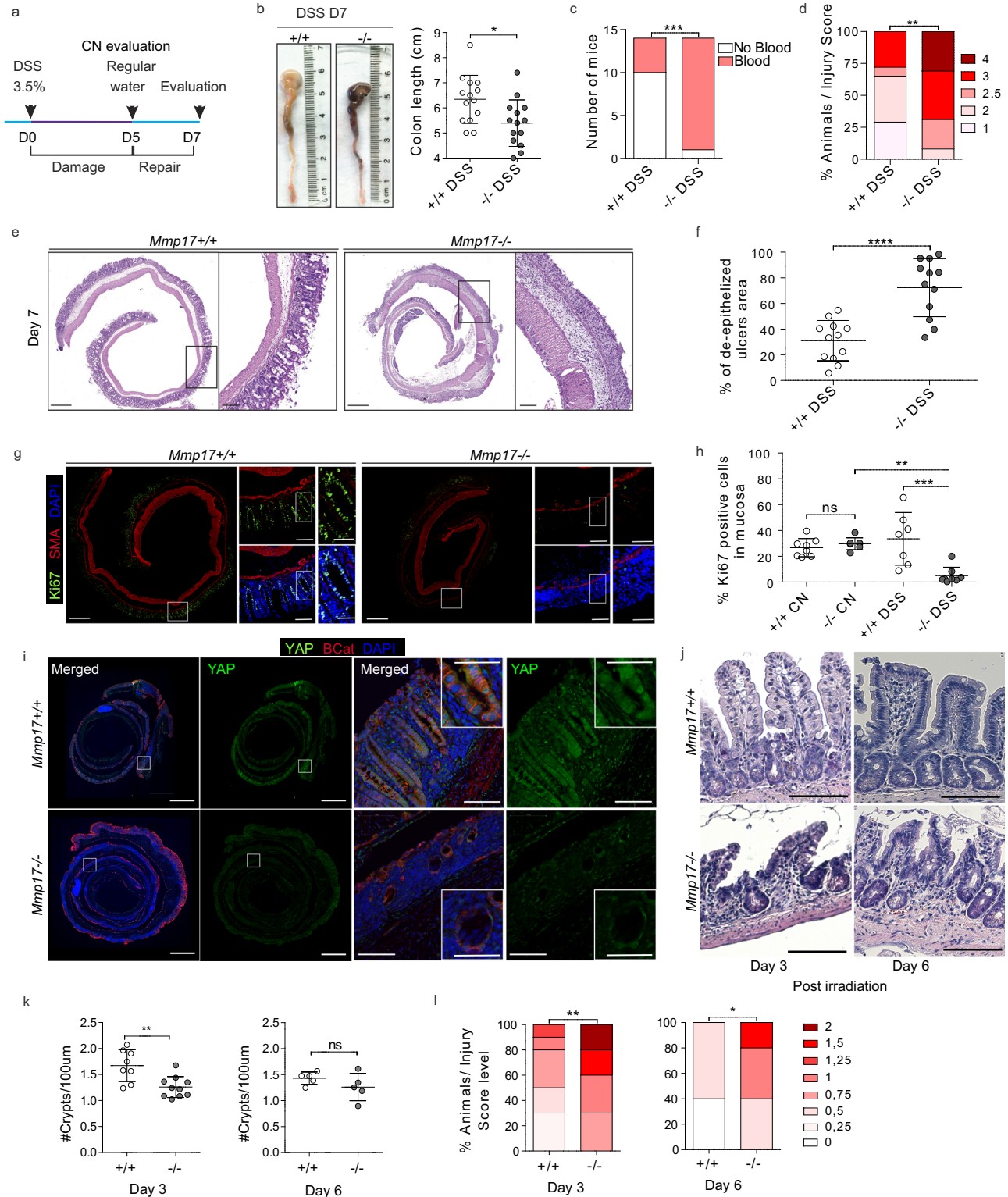

**PERIOSTIN is a muscle-derived factor cleaved by MMP17 and that can activate YAP and induce organoid growth**. MMP17 is expressed by intestinal smooth muscle cells and is required to allow for epithelial repair in vivo (Figs. 5, 6). Furthermore, muscle-SN is able to reprogram epithelium towards a reparative state in vitro and can stimulate de novo crypt formation (Fig. 7). We, therefore, hypothesize that MMP17 cleaves a smooth muscle-derived factor that facilitates epithelial reprogramming that is needed for repair. To identify said factor, we examined smooth

muscle-derived proteins by performing mass-spectrometry on muscle-SN, and we detected 550 proteins (Supplementary Data 2). We next curated this list to only display proteins known to be growth/niche factors and/or extracellular proteins (Fig. 8a). Among these was PERIOSTIN (POSTN), which is a matricellular protein that is highly expressed in smooth muscle (Fig. 8b). We previously identified a different matricellular protein, OSTEO-PONTIN (OPN), as an MMP17 substrate[24] (Fig. S8a). Of note, POSTN can serve as a ligand for ITGAV to activate AKT and

**Fig. 5 Muscle-specific Mmp17 is required for intestinal repair after injury. a** Timeline of DSS treatment. **b** Representative colonic images from DSS-treated animals. Graph shows colon length, and blood in stool and/or colonic cavity at D7 in (**c**). $n = 14$ from five independent experiments. **d** Histology-based blind scoring of DSS-derived injury in the colon. $n = 14$. **e** Images of H&E staining of the distal colon. Scale 500 μm; 100 μm in magnified. **f** Graph represents damaged mucosa as % of total length. $n = 12$. **g** Images of DSS-treated distal colon stained for Ki67 (Green), SMA (Red) and DAPI (Blue). Scale 500 μm; 100 (left) and 50 (right) μm. **h** Graph shows % of Ki67+ cells in mucosa, normalized to total mucosal cell number. $n = 7$ mice in KO CN and WT DSS and eight mice per genotype in the rest. **i** Representative images of DSS-treated distal colon stained for YAP (green), β-Cat (red), and DAPI (blue). Scale = 1 mm (tile scan), 100 and 50 μm (insets). $n = 3$. **j.** Representative images of H&E sections of small intestine 3 and 6 days after 10 Gy irradiation. Scale 100 μm. $n = 7$ WT, 8 KO analyzed from two independent experiments at Day 3 and $n = 5$ from Day 6. **k** Graph shows quantification of crypts/100 μm on Day3 and Day6 post irradiation. $n = 8$ WT, 10 KO at Day 3, and $n = 5$ at Day 6. **l** Blind scored injury level in irradiated mice. $n = 8$ WT, 10 KO (Day3) and $n = 5$ (Day 6). Numerical data in (**b**), (**f**), (**h**), and (**k**) are represented as means ± SD. Data in (**b**), (**f**) and (**k**) were tested by unpaired $t$-test (two-tailed) with $p = 0.0127$ in (**b**), $p < 0.0001$ in (**f**) and $p = 0.0033$ in (**k**). Data in (**c**) was analyzed by Fisher's exact test (One-tailed, $p = 0.0007$) and **d** and **l** were analyzed by Mann–Whitney test (two-sided) ($p = 0.0030$ in (**d**), $p = 0.0078$ and $p = 0.0434$ in day 3 and 6 in (**l**), respectively). One-way ANOVA ($F =$ ,10.01; $p = 0.0009$), was applied in (**h**), followed by Tukey's Multiple Comparison test ($-/-$CN vs $-/-$DSS **$p < 0.01$; $+/+$DSS vs $-/-$DSS ***$p < 0.001$). Source data are provided as a Source Data file.

YAP signaling[43–45]. Further, POSTN has been proposed to capture BMP members in the ECM[46]. These features may be relevant for biological processes we identified to be affected by MMP17. Co-incubation of human POSTN with MMP17 led to several POSTN fragments indicating MMP17 is able to cleave POSTN (Fig. 8c). In addition, we examined POSTN in MMP17-WT and KO smooth muscle and we found a decrease in POSTN fragments in KO muscle samples, suggesting that cleavage of POSTN also occurs in vivo and it is impaired in KO intestines (Fig. 8d). Next, we used mass spectrometry (MS) to identify putative cleavage sites in POSTN after incubation with MMP17. After tryptic digestion we found at least nine cleavage sites that had five or more peptide-spectrum matches and were not caused by tryptic digestion (Figs. 8e, S8b,c). Several of these could explain the presence of the ~25 kDa fragment we found in vivo as all identified cleavage sites are conserved in both mice and human. In order to define which of these cleavage sites could be targeted in vivo, we modeled in silico the interaction between MMP17 active form (membrane-bound MMP17 dimer) and POSTN. As shown in the 3D model of Fig. 8f (and Fig S8c) cleavage site 664 between I and P amino acids was the best-suited candidate, located at MMP17 active site. In addition, three other cleavage sites in the POSTN molecule, 157VN, 291MG, and 793GG, would be accessible for cleavage according to in silico docking. Of note, cleavage of 291MG would expose the Integrin alpha-5 binding motif of POSTN which is located within amino acids 300-314 in the FAS2 domain[47]. Finally, we found that POSTN itself modestly induces SI organoid growth that is associated with the induction of Ki67 and nuclear YAP (Fig. 8g–j).

## Discussion

The role of smooth muscle cells in the intestine, other than in peristalsis, has been largely undefined. However, in a recent preprint, it was found that in early human gut development ACTA2+ muscularis mucosa cells are the major source of WNT, RSPO, and GREM niche factors[48]. In contrast, in adult mice, there have been various mucosa-resident mesenchymal cell populations described that express *Rspo* and *Wnt* genes[10,12,13]. We here find that smooth muscle cells, and in particular, the ones residing in the muscularis mucosa, are the primary source of BMP antagonists such as *Grem1/2* (Fig. 1). The importance of GREMLIN1 as an ICS niche factor was recently determined by an experiment in which *Grem1*-expressing cells were depleted using the diphtheria-toxin system, which also led to the rapid loss of LGR5+ ISCs[8]. Based on our results, this experiment would thus have led to the death of practically all muscularis-mucosa smooth muscle cells. Our work here highlights the importance of smooth muscle for providing niche factors important for ISC homeostasis. Thus, future studies are warranted in which niche factors

are deleted in a cell-type specific manner to determine the relative importance of each cell type as a niche factor provider.

We find an important role for MMP17 in vivo. We show that KO mice have a reduction of ISC-associated genes in homeostasis (Fig. 4), which may be caused by increased levels of SMAD4 at the bottom of KO crypts (Fig. 3). Indeed, BMP signaling can impair ISC-signature genes by direct repression via SMAD1/4 recruitment of HDAC1[49]. We further identify that MMP17 is required for intestinal epithelial repair after injury, which we potentially link to the ability of MMP17 to cleave POSTN. We hypothesize that MMP17 cleavage of POSTN is necessary upon injury to reprogram epithelial cells in a YAP-dependent manner. Thus, cleavage of POSTN by MMP17 is an activation step in our model. However, there may be other smooth-muscle cell-derived factors in play as POSTN was only able to induce organoid area by 25% compared to double or triple the size by Muscle-SN, even though we observed YAP activation in both (Figs. 7c and 8j). Recently, Ma *et al.* described a role for POSTN in activating YAP/TAZ through an integrin-FAK-Src pathway using colon cancer cell lines[45]. In addition, it was found that POSTN can act through Integrin alpha-5 in various cell types[43,44]. We here extend these findings and show that POSTN can also affect primary non-tumor intestinal epithelial cells (Fig. 8). In addition to this direct role affecting the epithelium, others have shown that POSTN can alter the ECM by binding to a variety of ECM-associated proteins including BMP1, FIBRONECTIN, and TENASCIN-C[50]. We speculate that through this link with the ECM, cleaved POSTN may broadly affect intestinal homeostasis in vivo, which would be impaired in MMP17-deficient mice such as we observe. In addition, others have also shown that POSTN is cleaved for example by CATHEPSIN K which results in a 35–40 kDa product during bone remodeling[51]. Finally, *Postn*-deficient mice have general issues with repair in various tissues[52,53]. Thus, although the importance of cleavage of POSTN remains not fully comprehended[54], it is clear that POSTN plays an important role in reparative processes and the identification of MMP17 as an enzyme that can cleave POSTN can be used in future studies.

MMP17, unlike other MMP members, exert minimal activity against classical ECM components[55]. So far, only a few bona fide substrates have been described in other tissue environments, such as ADAMTS-4[56], alphaM integrin[57], and two matricellular proteins; OPN[24] and POSTN described in this work. We cannot discard the possibility that these or other unidentified MMP17 substrates play a role in ISCs niche regulation or the intestinal response to injury, particularly OPN which expression is increased in patients with inflammatory bowel disease[58]. Furthermore, in a tumor environment, both ADAMTS4 and OPN are overexpressed in colon cancers[59–61]. Finally, also non-

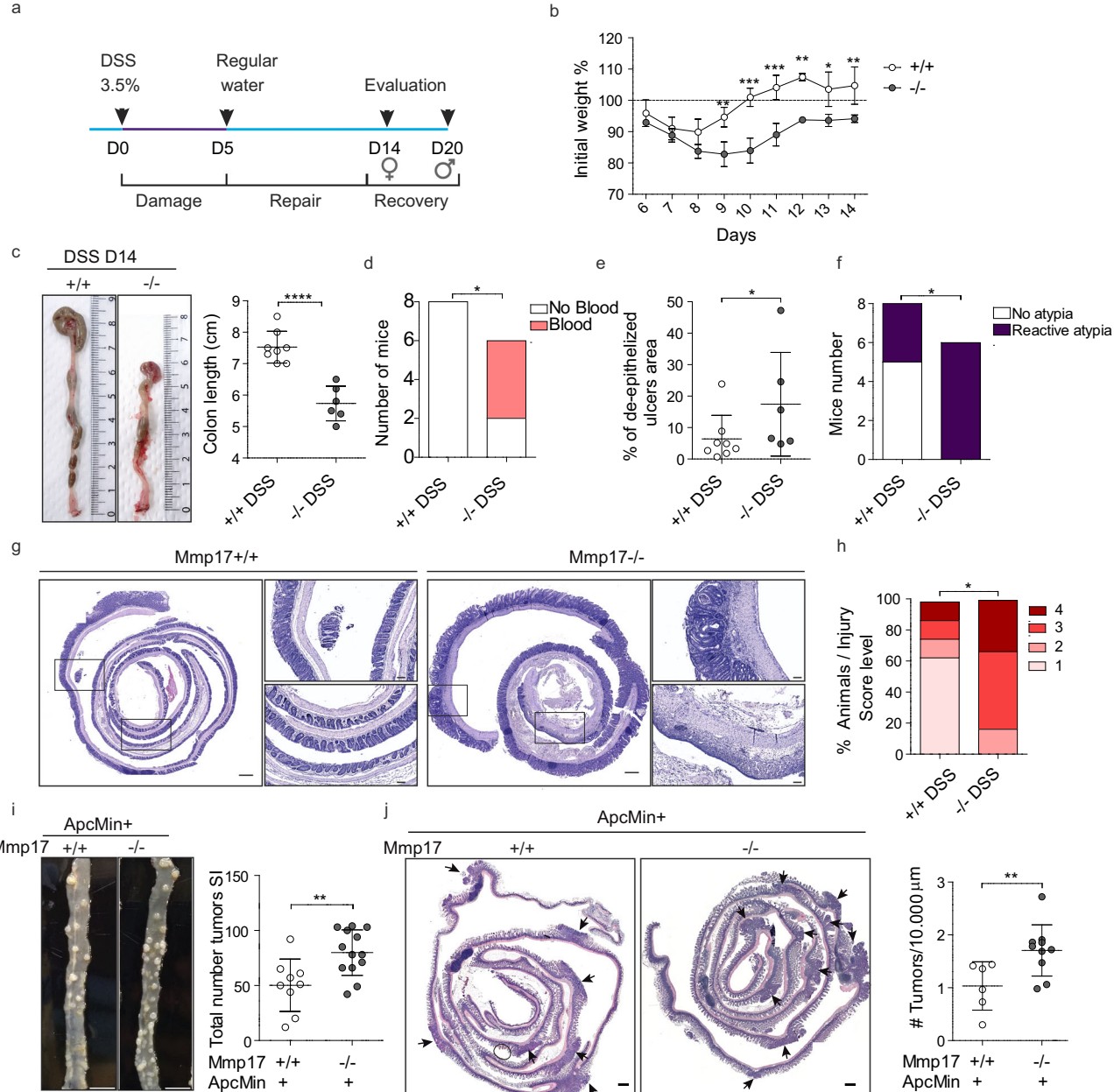

**Fig. 6 MMP17 absence hinders long-term repair in mouse intestinal epithelia and leads to increased tumorigenesis. a** Timeline of DSS-long treatment. Mice were exposed to 3.5% DSS for 5 days followed by 14 (females) or 19 days (males) days of regular water to allow epithelial restoration. **b** Weight loss relative to % to initial weight. $n = 3$ KO and 4 WT mice per genotype. Represented one experiment with females. **c** Representative images showing colon length at end point and its quantification. **d** Graph shows the presence of blood in stool/colon lumen at end point. **e** Graph represents the percentage of unhealed areas (ulcers) in WT and KO colonic mucosa at end point. **f** Graph shows the incidence of reactive atypia (or reactive epithelial changes after DSS). **g** Representative H&E pictures of colon swiss rolls showing healed crypts in WT vs ulcered areas and reactive atypia in KO. Scale 500 μm. **h** Injury score representation of damaged colon evaluation. $n = 6$ KO and 8 WT mice per genotype in (**c**–**h**). **i** Representative pictures showing tumor incidence in a portion of the small intestine of ApcMin + Mmp17 WT and KO mice (Jejunum). Scale bar 1 cm. Graph shows total number of tumors counted in fresh tissue (small intestine complete length). $n = 9$ WT and 13KO mice per genotype. **j** H&E representative images of small intestine swiss rolls in transverse cut. Arrows highlight visible tumors. Scale bar 500 μm. Graph shows tumor quantification normalized to tissue length. $n = 6$ WT and 10 KO mice per genotype. Numerical data in (**b**) was analyzed by two-way ANOVA ($F = 11.56$, ***$p < 0.001$), followed by Bonferroni post-test, asterisks correspond to comparison between WT and KO in each time point (*$p < 0.05$, **$p < 0.01$, ***$p < 0.001$). Fisher's exact test, one tailed was used in (**b**) and (**f**) ($p = 0.0150$ in (**d**) and $p = 0.0280$ in (**f**)). Numerical data in e and h were analyzed by Mann–Whitney $t$-test (two-sided) with $p = 0.0296$ in (**e**) and $p = 0.0230$ in (**h**). Data in (**c**), (**i**), (**j**) were analyzed by unpaired $t$-test (two-tailed, $p < 0.0001$ in (**c**), $p = 0.0055$ in (**i**) and $p = 0.0075$ in (**j**)). Source data are provided as a Source Data file.

catalytic activities for MMP17 have been described related to tumors[62].

To summarize, we discover a previously unappreciated role for intestinal smooth muscle tissue. We find that smooth muscle cells are likely the major contributors of BMP antagonists, which are essential niche factors for the maintenance of ISCs. In addition, we describe an important role for smooth-muscle restricted expression of MMP17, which in vivo is required for epithelial

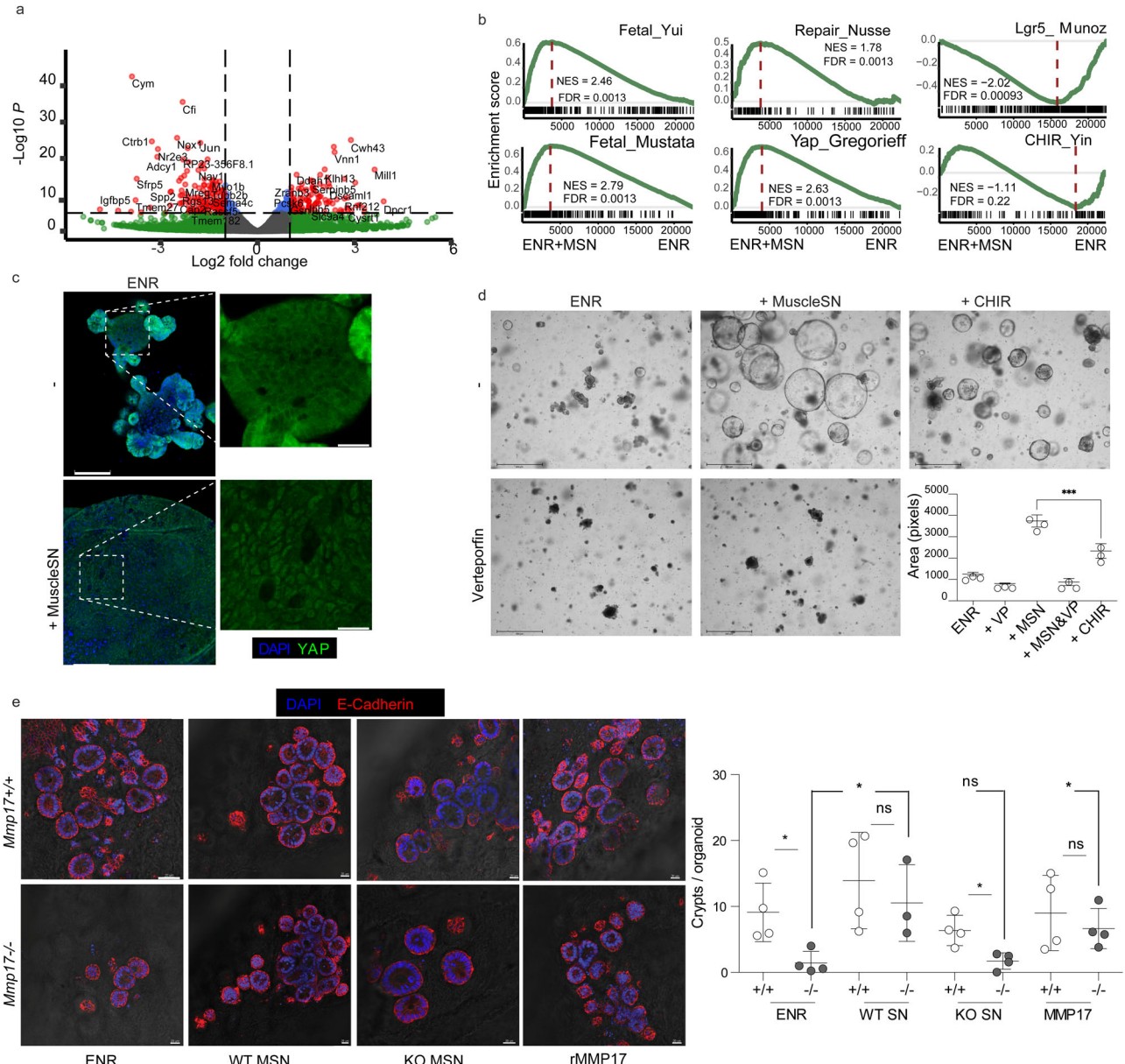

**Fig. 7 Muscle-SN induces a reparative epithelial state in organoids and rescues the reparative growth on KO iECM. a** Volcano plot showing significantly up and downregulated genes in SI organoids exposed to muscle-SN vs ENR. **b** GSEA of indicated gene sets comparing ENR with Muscle-SN (MSN) treated SI organoids. NES (normalized enrichment score) and FDR (false discovery rate) are depicted. **c** Representative maximal projection confocal images showing cytoplasmic vs nuclear YAP staining (green) in ENR vs Muscle-SN treated SI organoids. Scale 100 μm; insert is 25 μm. $n = 2$ independent experiments. **d** Representative images of WT SI organoids 3 days after splitting (scale is 650 μm), grown in the presence of indicated factors. Average area per organoid was quantified for three independent wells. **e** Representative immunofluorescent images of de novo crypt formation on WT and KO iECM with different treatment conditions. Matrigel-cultured 3-day old SI organoids were plated on WT and KO iECM. Adhered organoids were quantified at day 1 post-plating and de novo crypt formation per organoid was analyzed at day 7. Culture media was supplemented with either WT MSN, KO MSN, or rMMP17. Scale bar 40 μm. $n = 4$ samples per condition analyzed in two independent experiments. Numerical data are means ± SD and were tested by one-way ANOVA ($F = 98.8$; $p < 0.0001$) followed by Bonferoni post-test (MSN vs CHIR $p < 0.001$) in (**d**). Numerical data in (**e**) is average of crypts per iECM pooled from two independent experiments ($n = 4$ mice). One-way ANOVA ($F = 3.714$; $p = 0.0079$) and unpaired $t$-test to compare groups (*$p < 0.05$). Source data are provided as a Source Data file.

repair. Finally, we provide evidence that MMP17 may act via cleavage of the matricellular protein POSTN, which in itself can induce repair-like features in the intestinal epithelium.

## Methods

**Mice**. *Mmp17−/−* mice in the C57BL/6 background have been described previously[35]. Mice were handled under pathogen-free conditions in accordance with CoMed NTNU institutional guidelines. Experiments were performed

following Norwegian legislation on animal protection and were approved by the local governmental animal care committee. Particularly, experimental designs for DSS, irradiation procedures, and ApcMin mice colony handling and tumor evaluation, were approved in advance by Norwegian authorities as stated in FOTS protocols (11842, 15888, and 17072). Mice included in these protocols were carefully monitored daily or weekly to avoid situations of moderate to high pain and comply with ethical procedures established prior to experiment development in agreement with CoMed facility at NTNU. End point protocols were applied when needed. All mice were genotyped by PCR of earclip samples using the following primers: Mt4-mmp SK1 5′-TCAGACACAGCCAGATCAGG-3′ SK2

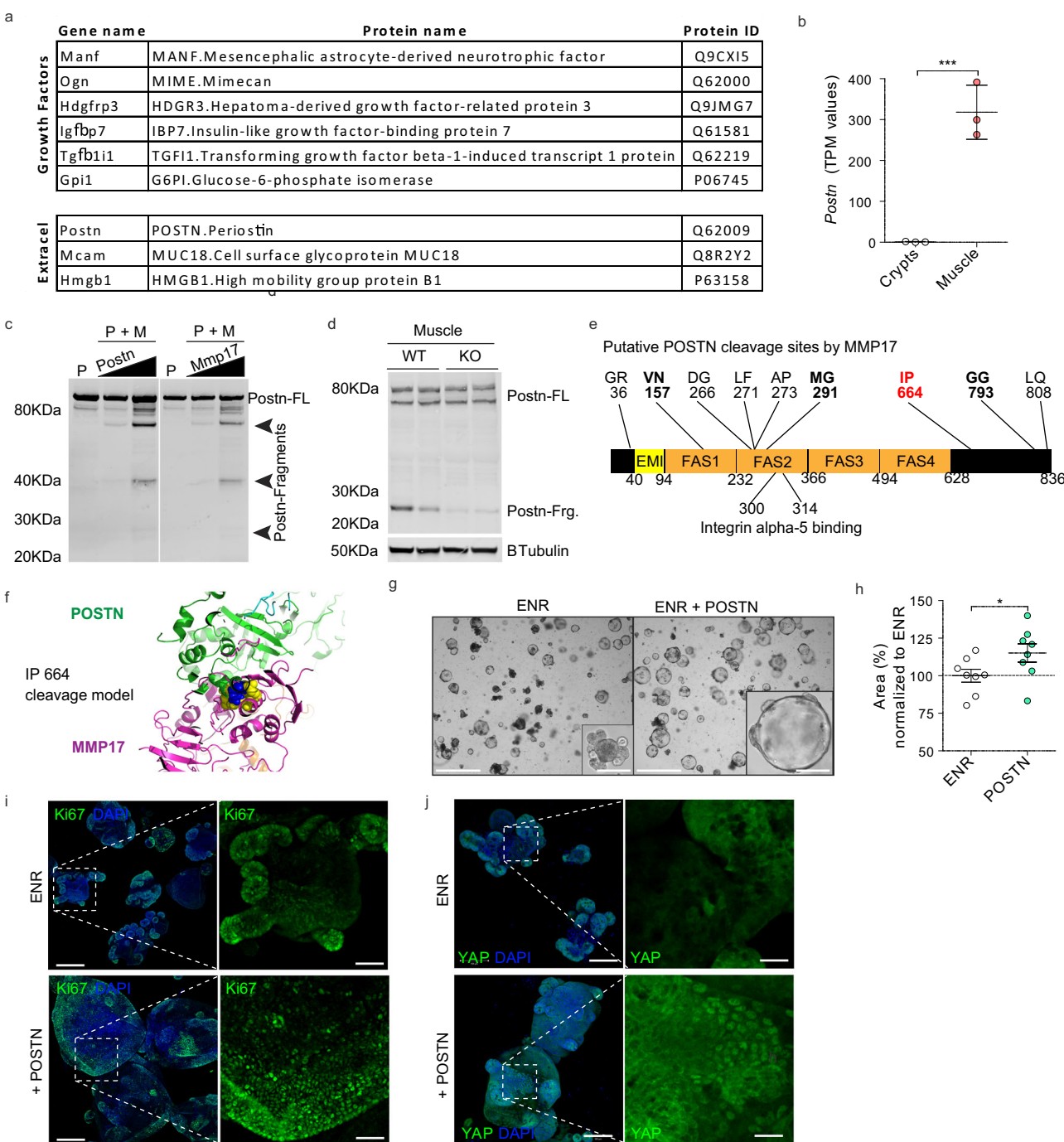

**Fig. 8 Identification of smooth muscle-derived factor PERIOSTIN as an in vivo and in vitro substrate for MMP17. a** List of growth factors and extracellular proteins found in muscle-SN. $n = 3$ biological replicates. **b** TPM values for *Postn* comparing crypts with smooth muscle tissue. $n = 3$ biological replicates. **c** In vitro digestion experiment with human recombinant proteins showing POSTN fragments when in contact with MMP17 catalytic domain. P, POSTN, M, MMP17, FL, full length. $n = 2$ experiments with increasing concentrations of POSTN or MMP17. **d** WB of mouse intestinal muscle showing decreased POSTN fragments in the absence of MMP17. $n = 2$ biological replicates. **e** Scheme of POSTN molecule showing putative sites of MMP17 cleavage based on digestion assays followed by MS. Sites that are noted all had 5 or more peptide-spectrum matches (PSMs). In blue and red are available sites after in silico modeling (see Fig. 8f for a model of the best fitted (red) site). **f** In silico model of MMP17 (magenta)- POSTN (green) docking showing close proximity of 664-665 POSTN cleavage site (cyan) to MMP17 catalytic site (yellow). **g, h** Representative brightfield images showing SI organoids morphology in the presence of POSTN and area quantification in (**h**) (Day 3). Scale 1250 and 200 μm in inset. $n = 8$ wells/genotype pooled from three independently performed experiments with 2–3 wells/experiment. **i, j** Representative confocal maximum intensity projection images showing YAP or Ki67 (green) staining in POSTN-treated SI organoids. Scale 100 μm; 25 μm in inset (YAP pictures) and 200 μm and 100 μm in Ki67 pictures. Numerical data are means ± SD. Data in (**b**) represents $p$-adjusted value from RNAseq analysis (padj value of 1.02e−06), and data in (**h**) was analyzed by Mann–Whitney test (two-sided, $p = 0.0260$). Source data are provided as a Source Data file.

5′- AGCAACACGGCATCCACTAC-3′ and SK3 5′-AATATGCGAAGTG-GACCTGG-3′ and ApcMin: 1. 5′-TTCCACTTTGGCATAAGGC-3′ 2. 5′-GCCATCCCTTCACGTTAG-3′ 3. 5′-TTCTGAGAAAGACAGAAGTTA-3′. Experiments were conducted on mice from 8 weeks to 20 weeks of age.

**Small intestine and colon crypt isolation.** Small intestinal crypts were isolated and cultured following previously published protocol to generate SI organoids[63]. Colonoids were cultured according to a published method[31]. Briefly, 10 first cm of the duodenum or the entire colon, were excised, flushed with cold PBS, and opened longitudinally. The internal surface of the duodenum was scrapped carefully with a coverslip to remove most of the mucus and part of the intestinal villus. Small pieces of duodenum or colon (2–4 mm in length) were cut with scissors and further washed with ice-cold PBS until the supernatant was clear. Next, tissue fragments were incubated in cold 2 mmol/L EDTA chelation buffer, for 30 min (small intestine) to 1 h (colon) at 4 °C. After removal of the EDTA buffer, tissue fragments were vigorously resuspended in cold PBS (small intestine) or cold chelation buffer (colon) using a 10-mL pipette-BSA coated to isolate intestinal crypts. The tissue fragments were allowed to settle down under normal gravity for 1 min, and the supernatant was removed for inspection by inverted microscopy. The resuspension/sedimentation procedure was repeated typically 8 times and the supernatants containing crypts (from wash 1 to 8) were collected in 50-mL Falcon tubes coated with BSA, through a 70 μm cells strainer to remove villi in case of the SI. Isolated crypts were pelleted, washed in PBS, and centrifuged at $200 \times g$ for 3 min at 4 °C to separate crypts from single cells. Crypts were resuspended in 10 ml basal crypt medium (BCM, advanced Dulbecco's modified Eagle medium -F12 supplemented with penicillin/streptomycin, 10 mM HEPES, 2 mM Glutamax) to quantify its number. After centrifugation and supernatant removal, crypts were resuspended in matrigel (Corning, 734-1101) and plated in P24 well plates (150 to 250 crypts/well) or 8-well ibidi imaging μsildes (80821, Ibidi). Crypts were cultured in ENR (BCM+ factors, explained below).

**Intestinal organoid culture.** SI organoids were cultured in ENR medium consisting of BCM (advanced Dulbecco's modified Eagle medium -F12 supplemented with penicillin/streptomycin, 10 mM HEPES, 2 mM Glutamax) + 1× N2 [ThermoFisher Scientific 100×, 17502048], 1× B-27 [ThermoFisher Scientific 50X, 17504044], and 1× N-acetyl-L-cysteine [Sigma, A7250]) and overlaid with ENR factors containing 50 ng/ml of murine EGF [Thermo Fisher Scientific, PMG8041], 20% R-Spondin-CM (conditioned medium, a kind gift from Calvin Kuo, Stanford University School of Medicine, Stanford, CA, USA), 10% Noggin-CM. For colonoids, same medium and reagents were use, plus 65% Wnt-CM (kind gifts from Hans Clevers, Hubrecht Institute, Utrecht, The Netherlands). We also made use of L-WRN (ATCC 3276) cell line conditioned media, which is considered as enriched media due to higher levels of Wnt, R-sponding, and Noggin than in the described ENR. Medium was renewed every other day. For passaging, organoids cultures were washed, and matrigel and organoids were disrupted mechanically by strong pipetting, centrifuged at $300 \times g$, 5 min at 4 °C and resuspended in matrigel to plate in P24 wells or eight-well ibidi chambers (80821, Ibidi). In different experiments SI organoids and/ or colonoids were exposed to ENR, ER, EN mediums, muscle-SN, obtained as explained below, human recombinant Periostin (50–500 ng) (3548-F2-050, R&D), CHIR99021 (3 μM, Sigma-Aldrich), or Verteporfin (3 μM, Sigma-Aldrich) for 3–5 days.

**Intestinal smooth muscle isolation and muscle-SN collection.** Smooth muscle samples were obtained as followed: after flushing with cold PBS, whole mice colon or 10 first cm of mice duodenum were open in longitudinal to expose the epithelial part and dissected under a bench scope. Using a coverslip, the epithelium and most of the mucosa (except the muscularis mucosae) were carefully removed under the dissection scope. The remaining tissue was further inspected under the microscope for residual crypts or fat and further cleaned. Samples were then fixed in PFA4% and stained as described below to ensure the presence of exclusively muscularis propria (circular and longitudinal layers) and some portions of muscularis mucosae and for the absence of epithelium. Smooth muscle samples were deep frozen in N2 (for RNA seq or WB analysis), rolled into swiss rolls for immunostaining, or cut in small pieces of muscle strips (around 2–4 mm long) used to obtain muscle-SN. To obtain this supernatant, the samples were then washed with sterile PBS twice, spin down, and collected in P24 well plates. One piece of muscle (2–4 mm) was included in each P24 well and cultured in 1 ml of DMEM-F12 for 24 h. Muscle-SN from different wells were then pooled and filtered (0,20 μm) and used directly + ENR factors in organoid cultures (right after splitting), sent for MS evaluation or frozen at −80 °C. Organoids were exposed to muscle-SN for 4–5 days. Muscle samples were obtained from the colon and used to obtain Muscle-SN, in RNA seq, WB or MS experiments. Small intestinal tissue was used for smooth muscle isolation for data in Fig. S2.

**Quantification and imaging of organoids.** P24 well plates were imaged using automated Z-stack in EVOS2 microscope with CO2, temperature, and humidity-controlled incubation chamber (Thermo-Fisher Scientific). SI organoid area and classification were evaluated using a custom analysis program written in python based on opencv2 (Lindholm et al., 2020). Images were autoscaled, and a canny edge detection algorithm was run on each individual z-plane using the cv2.canny function. Small pixel groups were removed and a minimal projection of the edges was generated. The contour of objects was defined based on this image. A watershed algorithm was used to split somewhat overlapping objects from each other and the object center was defined as the pixel furthest from the edge of the object. Each object was extracted in a 120×120 image on a white background and classified as either "Junk", "Budding" or "Spheroid" in a neural network implemented and trained using Tensorflow and Keras. A custom visual classification editor was then used to correct the locations of organoid centers and the classification group. The order of images was randomized, and treatment information hidden during the classification correction. The images then went through a second segmentation step where the segmentation was re-done as described above. The objects of contours with multiple centers were then split apart with a watershed algorithm that used the corrected organoid centers to split overlapping organoids. Organoids formation efficiency in SI organoids and in colonoids was calculated by manually counting the number of successful organoids 24, 48, and 72 h after crypt isolation in EVOS2 bright field pictures. SI organoids size was evaluated from day 1 to 5. For muscle-SN experiments quantification was performed at day 4 and at day 3 in POSTN treatments.

**Immunofluorescence staining in organoids and imaging.** For immuno-fluorescent labeling and imaging, SI organoids were grown in 70% Matrigel-30% ENR on eight-chamber Ibidi μslides (80821, Ibidi). SI organoids were fixed in PBS containing 4% paraformaldehyde (pH 7.4) and 2% sucrose for 30–45 min, permeabilized, and blocked in PBS-Triton X-100 0.2%, 2% normal goat serum (NGS), 1% BSA Glycine 100 μM for 1 h at RT. Next, SI organoids were incubated with primary antibodies against the following antigens diluted in PBS-TX100 0.2% + BSA 0.5% + NGS 1%: Ki67 (1:200, rabbit monoclonal antibody (mAb), Invitrogen, MA5-14520), β-catenin (1:200, mouse mAb, BD Biosciences, 610154), and YAP (1:100, rabbit mAb, Cell Signaling, 14074) overnight at 4 °C with slow agitation. Organoids were washed in PBS containing 0.1% Tween20 and incubated overnight in the same buffer at 4 °C with the appropriate Alexa Fluor secondary antibody (1:500) along with Hoechst 33342 (1:10,000). SI organoids were washed with PBS buffer with 0.1% Tween 20 and mounted using Fluoromount G (ThermoFisher Scientific, 00-4958-02). SI organoids were imaged in a Zeiss Airyscan confocal microscope, using a 10x and 20x objective lens. Images were analyzed using Zen black edition software (Zeiss) and maximal projections are shown. For immunofluorescent staining on re-epithelialized iECM—samples were fixed with 4% PFA for 20 min, and blocked with blocking buffer (5% Horse serum, 0.2% BSA, and 0.3% Triton X-100 in PBS) for 30 min at room temperature. Samples were then incubated with Anti-E-Cadherin antibody (DECMA-1 clone, Sigma, MABT26) and DAPI (1 μg/ml, Sigma) for 48 h at +4 °C on a shaker. Following the washes (3×) with the blocking buffer, samples were incubated with Alexa Fluor secondary antibody overnight at +4 °C. Samples were incubated with 80% glycerol in PBS overnight before imaging with Leica TCS SP8 STED confocal microscope.

**RNA seq of organoids and tissue.** Pieces of clean colon muscle, colon crypts (obtained using colon crypt isolation protocol previously described), or pooled SI organoids were used for RNA isolation. Tissues were first placed in lysis buffer (RLT, provided in RNeasy® Mini Quiagen Kit, 74104) and disrupted with sterile ceramic beads (Magna Lyser green beads tubes 03358941001) using a Tissue Lyser (FastPrep-24™, SKU 116004500), with two rounds of 6500 rpm for 30 s each, with care taken to maintain the sample cold. SI organoid wells were first disrupted by strong pipetting and pelleted before lysis. RNA isolation was performed following manufacturer instructions (RNeasy® Mini Kit, Qiagen, 74104, for tissue and Direct-zol™ RNA MiniPrep, BioSite R-2052, for organoids). RNA was quantified by spectrophotometry (ND1000 Spectrophotometer, NanoDrop, Thermo Scientific) and 25 μl at a 50 ng/μl concentration of RNA were used for RNA seq. Library preparation and sequencing for tissue RNA seq was performed by NTNU Genomic Core facility. Lexogen SENSE mRNA library preparation kit was used to generate the library and samples were sequenced at $2 \times 75$ bp paired end using Illumina NS500 flow cells. Library preparation and sequencing for organoid RNA seq was performed by Novogene (UK) Co. NEB Next® Ultra™ RNA Library Prep Kit was used to generate the library and samples were sequenced at $150 \times 2$ bp paired end using a Novaseq 6000 (Illumina). The STAR aligner was used to align reads to the Mus musculus genome build mm10[64,65]. featureCounts was used to count the number of reads that uniquely aligned to the exon region of each gene in GEN-CODE annotation M18 of the mouse genome[66]. Genes that had a total count of <10 were filtered out. DESeq2 with default settings was used to do a differential expression analysis[67]. Heatmaps were generated using the R-package pheatmap[68]. PCA analysis was done with the scikit-learn package using the function sklearn.decomposition.PCA[69]. Gene set enrichment analysis (GSEA) was performed on the full list of genes from differential expression sorted by log2 fold change and with log2 fold change as weights. GSEA was run with the R package clusterProfiler using 10,000 permutations and otherwise default settings[70]. A list of the top 250 genes upregulated in crypts was used for Enrichr analysis[36,37].

All raw sequencing data are available online through ArrayExpress: WT and KO smooth muscle and crypt RNA seq: E-MTAB-9180;ENR vs MuscleSN treated SI organoids RNA seq: E-MTAB9181.

**Mass spectrometry of muscle-SN**. Muscle supernatants were collected as stated above. For each sample, 100 ml of soluble protein was taken and proteins were reduced with 4 mM DTT at room temperature for 1 h, and alkylated with 8 mM iodoacetamide at room temperature for 30 min in the dark, after which additional 4 mM DTT was added. A first digestion was carried out with 40 ng Lys-C at 37 °C for 4 h. The samples were diluted four times and further digested with 40 ng trypsin at 37 °C overnight. Digested protein were desalted using Sep-Pak C18 cartridges (Waters), dried by vacuumcentrifuge and stored at −20 °C for further use.

For MS analysis the peptides were dissolved in 2% formic acid and 5 μl of each digested sample was injected on an UHPLC 1290 system (Agilent) connected to an Orbitrap Q Exactive HF spectrometer (Thermo Scientific). Reconstituted peptides were trapped on an in-house packed, double-fritted (Dr Maisch Reprosil C18, 3 μm, 2 cm × 100 μm) precolumn for 5 min in solvent A (0.1% formic acid in water) at 5 μl/min before being separated on an analytical column (Agilent Poroshell, EC-C18, 2.7 μm, 50 cm × 75 μm). Solvent A consisted of 0.1% formic acid, solvent B of 0.1 % formic acid in 80% acetonitrile. Separation was performed at a column flow rate of ~300 nl/min (split flow from 0.2 ml/min) using a 95 min gradient of 13–44 % buffer B followed by 44–100% B in 3 min and 100% B for 1 min was applied. MS data were obtained in data-dependent acquisition mode. Full scan MS spectra from $m/z$ 375–1600 were acquired at a resolution of 60,000 to a target value of $3 \times 10^6$ or a maximum injection time of 20 ms. The top 15 most intense precursors with a charge state of 2+ to 5+ were chosen for fragmentation. HCD fragmentation was performed at 27 normalized collision energy on selected precursors with 16 s dynamic exclusion at a 1.4 $m/z$ isolation window after accumulation to $1 \times 10^5$ ions or a maximum injection time of 50 ms. Tandem mass spectrometry (MS/MS) spectra were acquired at a resolution of 30,000.

Proteins IDs and intensity values (abundance) are represented in Supplementary Data 1.

**MS data analysis**. All raw MS files were searched using MaxQuant software (version 1.6.10.43). MS/MS spectra were searched by Andromeda against a reviewed Uniprot Mus Musculus (17,068 entries, 2020) using the following parameters: trypsin digestion; maximum of three missed cleavages; cysteine carbamidomethylation as fixed modification; oxidized methionine and protein N-terminal acetylation as variable modifications. Mass tolerance was set to 20 ppm for MS1 and MS2. The protein and PSM False Discovery Rate (FDR) were set to 1%.

**Recombinant MMP17-Periostin cleavage site discovery**. Recombinant Human periostin (3548-F2-050 R&D systems) and MMP17 (human) recombinant (P4928 Abnova) were dissolved in 50 mM Tris-HCL, 10 mMCacl$_2$, 80 mM NaCl pH7.4 to reach a concentration of 250 ng/μl. 18 μl MMP17 and 72 μl Periostin were mixed and incubated for 2 h at 37 °C. After incubation 22.5 μl of sample buffer with DTT (XT sample buffer 4x Bio-Rad) was added. Samples were heated for 5 min 95 °C, before loading on a 12% gel (Criterion, Bio-Rad). The gel was run for ~3 cm, then stopped and stained with Imperial safe stain (Pierce, Thermo). The gel lane was cut in 3 pieces and the excised gel pieces were reduced with DTT, alkylated with iodoacetamide, and in-gel digested with trypsin[71].

Samples were analyzed with LC-MS performed on an UltiMate 3000 RSLCnano System (Thermo Scientific) coupled to a Orbitrap Exploris (Thermo Scientific). Peptides were first trapped onto an in-house packed precolumn (3 μm C18 Dr. Maisch ReproSil, 2 cm × 100 μm) and eluted for separation onto an analytical column (Agilent Poroshell, EC-C18, 2.7 μm, 50 cm × 75 μm). Solvent A consisted of 0.1% formic acid, solvent B of 0.1 % formic acid in 80% acetonitrile. Trapping was performed for 5 min at 5 μL/min flowrate in solvent A. Peptides were separated at a column flowrate of 300 nl/min using a 95 min gradient of 12–44 % buffer B, followed by 30–100% B in 3 min, 100% B for 1 min.

The mass spectrometer was operated in data-dependent acquisition (DDA) mode. Full scan MS was acquired from 375–1600 $m/z$ with a 60,000 resolution at 400 $m/z$ and an accumulation target value of 3e6 ions. Up to 15 most intense precursor ions were selected for HCD fragmentation performed at normalized collision energy (NCE) 27, after accumulation to target value of 1E5. MS/MS acquisition was performed at a resolution of 30,000.

Proteome data (RAW files) were analyzed by Proteome Discoverer (version 2.4.1.15, Thermo Scientific) using Percolator and standard settings. MS/MS data were searched against the Swissprot database (564277 entries, 01-2021) with Mascot. The maximum allowed precursor mass tolerance was 10 ppm and 0.05 Da for fragment ion masses. False discovery rates for peptide and protein identification were set to 5%. Trypsin was chosen as cleavage specificity allowing two missed cleavages. Carbamidomethylation (C) was set as a fixed modification, while oxidation (M) and acetyl (Protein N-term) were used as variable modifications.

**Tissue immunostaining and immunohistochemistry**. Intestines were open in longitudinal, flushed, and rolled into "Swiss rolls" as previously described[72]. Tissues were then fixed in PFA 4% 16 h and, in the case of frozen samples, included in sucrose 2% from 3 h to over night. Swiss rolls were then included in paraffin or in OCT compound for frozen sectioning. Transversal cuts of swiss rolls were used in all experiments. For immunohistochemistry and H&E staining, paraffin sections of 4 um were subjected to normal deparaffination and hydration. Antigen retrieval protocol (citrate buffer or pH9 EDTA buffer) was used for immunohistochemistry.

After blocking unspecific binding on BSA 2%, NGS 5% PBS-TritonX100 0.3%, samples were incubated overnight with one of the following primary antibodies: anti-Muc2 (1:200, Santa Cruz, rabbit mAb, sc-515032) or anti-Lysozyme (1:500, rabbit polyclonal antibody (pAb), Dako, A0099) in same blocking buffer. Specific secondary antibody and DAB (Vector technology) protocol were used. For LacZ staining, frozen sections were stained following β-gal staining kit indications (K1465-01, Fisher Scientific). Tissue sections were imaged using brightfield microscope EVOS2 with 10 or 20× objective lens and tile scan for the visualization of the complete swiss roll was performed when needed. For immunofluorescence, sections of 4–7 μm were incubated in blocking buffer for 1 h (PBS-Tx100 0.3%, NGS 5%, BSA 2%, Glycine 100 μM), followed by primary antibody incubation overnight. The following antibodies were used for immunofluorescence: anti-βGalactosidase (βGal, Frozen sections, Dil 1:100, Rabbit polyclonal antibody, Abcam, ab4761), anti-Ki67 (Paraffin, 1:200, rabbit monoclonal antibody, Invitrogen, MA5-14520) anti-SMAD4 (Frozen sections, methanol 10 min −20C, Dil 1:400, Cell signaling, 46535), anti-βcatenin (Paraffin, 1:200, Mouse mAb, BD Biosciences, 610154), anti-Olfm4 (Paraffin, 1:200, Rabbit mAb, Cell signaling, 39141),, anti-cleaved caspase 3 (Paraffin, Dil 1:200, Rb pAb, Cell signaling, 9661), anti-pSMAD1/5/9 (Frozen sections, 10 μm, Dil 1:800, 13820, Cell Signaling, following TSA amplification, NEL744001KT, Perkin Elmer, following previous specifications[73]), anti-YAP (Paraffin, 1:100, rabbit mAb, Cell Signaling, 14074), anti-Desmin and anti-CD31 (Frozen sections, Dil 1:200 for both antibodies, Mouse mAb, Thermo Fisher Scientific,MA5-13259 and Hamster mAb, Millipore, MAB1398z, respectively) and anti-CD45 (Frozen sections, 1:200, rat mAb, Abcam, 25386), and anti E-Cadherin (Rabbit mAb, Cell signaling 3195). After washing the slides in PBS-Tween 20 0.2%, tissues were incubated with the appropriate Alexa Fluor secondary antibody, Anti-SMA-Cy3 directly labeled antibody (1:500, mouse mAb, Sigma-Aldrich C6198) and Hoechst 33342 (1:10,000). Tissue sections were imaged with a Zeiss Airyscan confocal microscope, using a 10× and 20× objective lens. Images were analyzed using Zen black edition software (Zeiss) and maximal projections are shown except for SMAD4 staining (single plane). Tile scans of swiss rolls were performed when required. The percentage of βGal positive cells in the muscle was calculated manually in Zen software by counting the total number of nuclei in muscularis mucosa or circular/longitudinal muscle vs βGal positive nuclei in these tissues.

**DSS colon injury**. Experimental epithelial colon injury was induced in 8–10 weeks mice by supplying 3.5% DSS (MP Biomedicals, 0216011010) in drinking water ad libitum during 5 days. Before (Day 0) and during experimental development (up to day 7 in short protocol or up to 19 days in long-term protocol), mice were monitored daily for signs of stress, pain, body weight loss, and presence of blood in stool (detected by hemoFEC, Cobas, 10243744). At day 5, DSS was replaced by regular water to allow epithelial renewal. Control mice to evaluate DSS damage level were euthanized at Day5. For the long-term protocol, weight recovery determined end point (14 days for females and 19 days for males). Colon tissue was then harvested, measured, imaged, fluxed with cold PBS, open in longitudinal, and processed for paraffin or OCT, as indicated above. Paraffin sections were stained for H&E, or Ki67 and SMA. H&E sections were imaged on EVOS2 (tile scan of complete swiss rolls) and genotype-blind analyzed for signs of injury. Crypt loss, presence of ulcered tissue in the mucosa, signs of immune infiltrate in mucosa and edema were evaluated to create an injury level profile for each sample (Injury Score). The % of de-epithelialized mucosa was calculated using Fiji (Image J) as the total length of intestinal surface devoid of epithelium vs total length of swiss roll. The % of Ki67+ cells were obtained by quantifying Ki67 positive nuclei in mucosa vs total nuclei in mucosa using Cell Profiler. Long-term DSS samples were evaluated by a pathologist to determine the presence of reactive atypia.

**Irradiation**. Eight-week-old male mice were subjected to whole body 10 Gy irradiation under anesthesia. Weight loss was evaluated at Day 0 and during the length of the experiment (3 or 6 days). Animals were carefully checked daily for signs of pain or stress. At end point, mice were euthanized and small intestine and colon were harvested, flushed with cold PBS, and processed in swiss rolls for paraffin. H&E sections were imaged and evaluated blindly for signs of impaired recovery of the mucosal tissue. Parameters as crypt loss, immune infiltrate, crypt length and villus length and width were evaluated to rate the injury score. Villus length and width were measured using Fiji (Image J). We evaluated caspase 3 levels through staining at Day 1 as a positive control for irradiation. Microcolony assay was performed on swiss rolls H&E images of Day3 and Day6 irradiated intestines and represented as #Crypts/100 μm of tissue. On average, crypts number was evaluated in 0.6–1 cm of ileum swiss roll extension in transversal cut, and more than 100 crypts counted/sample. Image quantification was performed using Fiji (Image J).

**Mmp17-ApcMin tumor evaluation**. The presence and number of intestinal tumors were evaluated in 18–20 weeks old $ApcMin + Mmp17+/+$, $Mmp17-/-$ or $Mmp17±$ (females and males) mice. Mice were weekly monitored for weight loss and signs of pain, to avoid situations of moderate to high pain. At end point, mice were euthanized and small intestinal tumors were visualized and counted. Brightfield pictures were taken to double check tumor number and quantify tumor size (Fiji, Image J). Tissues were fixed as swiss rolls overnight and processed in

paraffin or for OCT as described above 5 µm microtome cuts were stained for H&E and/or for immunostaining. Immunostaining was performed as previously described in paraffin for βCatenin and Olfm4 staining. For the evaluation of *Mmp17* expression in tumors, *Mmp17±* tissues were processed in OCT and stained for βGal as stated above.

**Protein extraction and western blot analysis**. Intestinal tissue (muscle or mucosa tissue isolated as previously described) was disrupted using a tissue lyzer and ceramic beads (Magna Lyser green beads tubes 03358941001) in cold regular RIPA buffer (+ protease and phosphatase inhibitors). Proteins (20–100 µg) were resolved by 7–12% SDS-PAGE in reducing conditions and transferred onto nitrocellulose membranes using iBlot2™ dry blotting technology (Thermo Fisher Scientific). Antibodies against SMAD4 (rabbit mAb, Cell signaling, 46535), POSTN (mouse mAb, SAB4200197, MERK, Sigma-Aldrich), β-tubulin (ab6160, Abcam,) and GAPDH (mouse mAb, Abcam, ab125247) were used at 1:1000 dilution (4 °C overnight). Immunoreactive proteins were visualized with corresponding fluorochrome-conjugated secondary antibodies (680 or 800 ODYSSEY IRDye®) and recorded by Licor Odyssey technology. Quantification of western blot bands was performed using ODYSSEY software (LI-COR Biosciences) and normalized to GAPDH or β-tubulin levels.

**In vitro digestion and western blot**. Human recombinant POSTN (250 ng to 1 µg; hrPOSTN, 3548-F2-050, R&D) or human recombinant OPN as control (1 and 2 µg; hrOPN, 1433-OP, R&D) were incubated in digestion buffer (50 mM Tris-HCl, 10 mM CaCl2, 80 mM NaCl, [pH7.4]) with or without human recombinant MMP17 (250 ng) (hrMMP17, P4928, Abnova) catalytic domain for 2 h at 37 °C. Samples were separated by 12% SDS-PAGE and transferred to nitrocellulose membranes. Full-length hrPOSTN and fragments were detected with an anti-POSTN monoclonal mouse antibody (SAB4200197, MERK, Sigma-Aldrich). Full length and Ct fragments of OPN were detected using 1H3F7 mouse monoclonal antibody as previously described[24]. An anti-mouse secondary antibody was used to visualize the different protein sizes.

**In situ hybridization (ISH) of intestinal tissues**. ISH using RNAscope technology (ACD, Bio-Techne) was performed using freshly cut paraffin sections of intestinal swiss rolls of *Mmp17+/+* and *Mmp17−/−* mice. Chromogenic RNA-scope of *Lgr5* (Mm-Lgr5, 312171) and *Olfm4* (Mm-Olfm4, 311831) in colon and small intestine was performed following manufacturer indications (RNAscope® 2.5 HD Assay-BROWN). Sections were imaged using bright field automatized microscope EVOS2 (tile scan). Quantification of positive Lgr5 cells was performed using Fiji (Image J); total area of positive Lgr5 crypts was normalized to tissue length. Immunofluorescence RNAscope (FISH) was performed following manufacturer protocol (RNAscope® Multiplex Fluorescent v2) and the following probes were used: *Mmp17* (Mm-Mmp17-C4 ACD design of NM_011846.5), *Grem1* (Mm-Grem1-C3, 314741), *Grem2* (473981-C2), *Chrdl1* (Mm-Chrdl1, 442811). Zeiss Airyscan scope images are maximal projections of the different channels taken with 10× and 20× objectives.

**iECM-based crypt regeneration assay**. iECM was prepared as previously described[39]. To obtain a decellularized small iECM from the intestine, 10 cm of MMP17 WT or KO ileum was flushed with cold PBS and incubated overnight in MQ water. Dead cells were cleared by flushing the intestinal lumen with cold MQ water. Next, the intestine was cut into 1 cm pieces, followed by 3 h incubation with 1% Sodium Deoxy Cholate (Sigma) on a shaker (10 rpm) at RT. Tissue pieces were washed with MQ water for 10 min., followed by 2 h RT incubation with 1 M Sodium Chloride and DNaseI (1 U/10 µl, on a shaker). Tissue pieces were then cut open longitudinally and placed on 35 mm glassbottom dishes (MatTek) keeping the luminal side upward. Luminal orientation of the iECM was confirmed with the bright field microscope. The iECM was then used to assess crypts formation from organoids laying on top of this decellularized tissue. 3-day old Matrigel-cultured organoid were washed with cold PBS (3×) and plated on iECM. The number of adhered organoids on iECM was quantified under light microsocope after 24 h, and further cultured for 7 days in standard ENR media or in supplementation with WT/KO MSN or hrMMP17. At day 7, re-epithelialized iECM were imaged and number of crypts attached to the iECM were quantified (ratio Crypts/organoids). iECM was then fixed in 4% PFA for further IF analysis.

**In silico modeling of MMP17-POSTN docking and cleavage**. Fasta sequences of mature human POSTN protein (Q15063 with the signal-peptide removed) was aligned to Uniclust·30 (release 08-2018) and pdb70 (release 09-2019) databases using the hhblits/hhsearch tools of the HH-suite3[74] to obtain a multisequence alignment (MSA) and pdb templates for comparative modeling.

For docking the full dimer a comparative modeling was made using de RosettaCM[75] tool of the of Rosetta suite v3.12 (www.rosettacommons.org) with the fasta sequence, the alignment and templates obtained before.

The model with best structural alignment to the template (5yjg) and minimal energy was selected as final candidate. A final cycle of refinement for minimize clashes and energy was made with the relax tool[76] of Rosetta suite v3.12 (www.rosettacommons.org). This tool recomputes the side-chain coordinates of

the protein residues with account of the composition of the protein. As before, the models with the best energy and correct folding were selected as the final model.

Previous results for in vitro experiments show the putative cleavage sites in POSTN by MMP17 protein. The 271LF and 273AP sites are buried in the dimeric model.To evaluate the accessibly of the exposed residues, the model of POSTN was manually located close to the active-site of MMP17 modeled before[24] using the pymol 2.4 software (www.pymol.org) for each exposed position. The positions at 36GR and 266DG have strong steric impediment that makes access impossible.

In each case, the complex was modeled using the docking protocol of the Rosetta software suite v3.12 (www.rosettacommons.org)[77]. Models were clustered and manually analyzed in pymol to select the best candidate in each case. No model with the 808LQ position near to the active site of MT4-MMP was obtained.

Models for position 157VN, 291MG, and 793GG are near to the active site of MMP17 and, probably, are accessible. The 664IP position is in the active-site of MMP17 and is the best candidate.

**Statistical analysis**. The statistical analysis performed in each case is explained in detail in the corresponding figure legend, together with the n and the times the experiment was performed. Data were analyzed by two-tailed Student's *t*-test or by one- or two-way ANOVA followed by Bonferroni post-test for data with Gaussian normal distribution. Non-normal distributions were analyzed by Mann–Whitney test. The one-tailed Fisher's exact test was used to analyze the incidence of blood in stool or the presence of reactive atypia. Statistical tests were conducted with Prism 5 software (GraphPad Software, Inc.). Data are presented as mean ± S.D. and differences were considered statistically significant at $p < 0.05$: *$p$ value < 0.05, **$p$ value < 0.01, ***$p$ value < 0.001 and $p$ value < 0.0001.

**Reporting summary**. Further information on research design is available in the Nature Research Reporting Summary linked to this article.

## Data availability

All raw sequencing data are available through ArrayExpress: WT and KO smooth muscle and crypt RNA seq: The WT and KO smooth muscle and crypt RNA seq data generated in this study have been deposited in the ArrayExpress database under accession code "E-MTAB-9180". The ENR vs MuscleSN treated organoids RNA seq data generated in this study has been deposited in the ArrayExpress database under accession code "E-MTAB-9181". The mass spectrometry proteomics data have been deposited to the ProteomeXchange Consortium via the PRIDE[78] partner repository with the dataset identifiers "PXD020561 (MSN supernatant)" and "PXD025770 (POSTN cleavage)". Supplementary Data 2 with proteomic data generated in this study is provided in the Supplementary Information. All other relevant data supporting the key findings of this study are available within the article and its Supplementary Information files or from the corresponding author upon reasonable request. All source data are available as a Source Data file. Source data are provided with this paper.

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

## Acknowledgements
We would like to thank Dr. Motoharu Seiki for kindly sharing the *Mmp17*KO/KI mouse line and Dr. Kaisa Lehti for first inspiring this study. We would like to thank to Anne Beate L. Marthinsen for performing the irradiation, and the Department of Radiology and Nuclear Medicine (St. Olav's Hospital) for allowing the use of their instruments. We also thank Arne Wibe and Elin Rønne for their evaluation of colon crypt reactive atypia. We thank Shreya Gopalakrishnan for sharing reagents and input on organoids growth, Yashwanth Subbannayya and Time Veth for input on MS analysis, and Rosalie Zwiggelaar for assisting in mouse experiments. We thank the imaging (CMIC) and animal care (CoMed) core facilities (NTNU), as well as the histology and animal facilities at CNIC for assisting in this work. The WT KO crypt-muscle RNA-seq was done by the Genomics Core Facility at NTNU, which receives funding from the Faculty of Medicine and Health Sciences and Central Norway Regional Health Authority. This research was part of the Netherlands X-omics Initiative and partially funded by NWO (project 184.034.019). This work was further financially supported by the Norwegian Research Council (Centre of Excellence grant 223255/F50, and 'Young Research Talent' 274760 to M.J.O.) and the Norwegian Cancer Society (182767 to M.J.O.). MMA is the recipient of a Marie Skłodowska-Curie IF (DLV-794391).

## Author contributions
M.M.A. and M.J.O. designed the research project with help from F.M., M.A., P.K., and A.G.A. for specific experiments. M.M.A., S.I., P.M.V., H.T.L., M.J.D., F.M., S.H., A.D.S., M.A., P.K., A.G.A., and MJO planned and performed experiments and/or analyzed data. M.M.A. and M.J.O. drafted the manuscript and all authors provided input.

## Competing interests
The authors declare no competing interests.

## Additional information

**Peer review information** *Nature Communications* thanks Kari Basso, Michael Blennerhassett and the other anonymous reviewer(s) for their contribution to the peer review this work. Peer reviewer reports are available.

