## [Peer Review File · Nature Communications]

Reviewers' Comments:

Reviewer #1:

Remarks to the Author:

Epithelial-mesenchymal interactions are important in the development of the intestinal mucosa, and continue to be essential in post-natal life. Mesenchymal cells such as myofibroblasts and teleocytes (the "pericryptal stroma") are the sources of growth-regulatory factors for intestinal stem cells, such as the wnts that maintain the critical balance of wnt:BMP signaling. Here, the authors look at "smooth muscle" as an under-studied potential participant in this interaction. They conclude that smooth muscle express BMP antagonists and that culture supernatants can replace the BMP antagonist NOGGIN in organoid development in vitro.

They select MT4-MMP, known to be expressed in smooth muscle, for further study using a KO mouse. (global; not inducible). While MT4 absence did not detectably affect smooth muscle, there was increased SMAD4 signaling in the KO crypts, interpreted as an absence of BMP antagonists (such as NOGGIN) that arise from mesenchymal cells.

Since this was interpreted to suggest a role for smooth muscle-derived MT4 (?), they used the DSS model of colitis in control vs KO mice, showing a decrease in KI-67 labelling of proliferative epithelial cells in the KO animals, similarly, a radiation-induced injury model showed increased injury in H&E-stained sections of intestine, and there was increased tumour initiation in the APC/min x MT4 KO mouse model.

Returning to the supernatant/organoid model, they found that the supernatant may support organoid growth via YAP. Study of potential targets of MT4 identified periostin, and showed its decreased cleavage in the KO mice. Intact periostin was found to be a growth factor for organoids, but the mechanism was not identified, nor the effects of cleavage products. It is unclear if or why periostin needs to be affected by MT4, an effect lost in the KO mice.

--

While interesting, there are a number of major problems in the presentation and design of this work. The lack of consideration of the known roles of mesenchymal cells in interaction with epithelial stem cells is a major problem. Their contribution to the current study makes it difficult to interpret the present outcomes in either control or MT4-KO mice.

The techniques used are powerful but the cell biology and in vivo approaches have significant gaps, where there is no definitive outcome. This prevents the synthesis a clear picture of the multiple findings, each with interesting, but loosely defined significance. Thus, it is hard to see the proposed role(s) of a smooth muscle-derived enzyme (MT4/MMP17) along with expression of a diffusible factor (yet undefined) that is active in vitro.

The definition of "smooth muscle" is unclear throughout – it is probably an error to equate equally the 3 layers of longitudinal, circular and muscularis mucosa, especially. Technically, the method for obtaining smooth muscle (scraping) is known to remove villus tips, but to leave the lowest levels of the epithelium intact, thus allowing the myofibroblasts to continue to influence outcomes.

I did not see that the key expt of using supernatant from KO-derived smooth muscle to see if this tissue still expressed NOGGIN-like activity, and thus contributed to organoid development. Does the +/+ or -/- affect SMAD4 signaling in the organoids?

Can the various animal models be studied with better resolution so that a mechanism is discovered? These are otherwise just observational.

Practically, how does MT4 reach the stem cell niche? Was MT4 measured in the control case? Do the supernatants suppress SMAD4/BMP signaling in vitro? What mediates the smooth muscle-derived supportive effect on organoids?

The decellularized intestinal expts were not well enough defined to be useful.

Throughout, there is a tendency to overstate the significance of the findings, which is not supported by the data – eg, line 257, that MMP17 is required for epithelial reparative responses upon injury.

The title might need some thought for improved impact –what is an "intestinal ECM niche" ? and it is not clear that it is "define(d)..." by MMP17.

Reviewer #2:

Remarks to the Author:

General Comments

My review for the manuscript NCOMMS-20-27757 "Smooth muscle-specific MMP17 (MT4-MMP) defines the intestinal ECM niche" is focused on the mass spectrometry/proteomics contribution to the manuscript. The mass spectrometry data presented is adequate but more details in the method needs to be included and I have noted some discrepancies.

Specific Comments

1. The Methods and Materials need to be more specific and or edited.
 - a. Was the protein concentration measured after protein was extracted from the muscle for normalization? i.e. by Bradford or other type of protein concentration techniques?
 - b. On line 236 you describe a column that does not exist. I believe it should be a 50 mm length column, not 50 cm. And I don't believe this column comes in a 75 um ID (nano). Also you indicate an analytical column was used, please confirm – most proteomics is nano-LC. What was the flow rate of the LC? How much sample was injected?
 - c. There is no description on how the data was analyzed, what bioinformatic platform was used and what the search parameters were. How was a peak list generated? What database was searched? What was the FDR? What was the criteria to confidently assign a protein ID. This is a major omission.
2. Regarding Supplementary Table 1. What are columns S1, S2 and S3? I am assuming these are biological replicates of the smooth muscle? Or technical replicates? For the column "Unique peptides", is that a combination from the 3 apparent replicates?
3. On line 349 of the manuscript the authors indicate the list of proteins was curated to only display proteins known to be growth factors. I wanted to interrogate the data so I searched the Supplemental Table 1 for the proteins listed in Figure 7d and I only find Manf and Postn. I could not find any of the other listed proteins in Supplemental Table 1. Where did the Proteins listed in 7d come from?
4. I note that the Proteomic data has not been uploaded to a data depository.

Reviewer #3:

Remarks to the Author:

The manuscript addresses the contribution of MMP17+ cells within the intestinal muscle layers in regulating the intestinal stem cell niche during normal intestinal homeostasis and several different injury models in vivo and within organoid models. The authors provide several different lines of investigation that are initially focused on the role of BMP antagonists. Some of the MMP17-deficient data is quite striking in relation to changes to responses such as in repair after acute DSS injury. However other areas of the manuscript are less well developed, for example the radiation injury and APCMin tumor models. Rather than investigate these models in more detail, the authors switch to studying of periostin which is detected in muscle-SN by mass spec and its activity. However, beyond showing recombinant periostin induces altered duodenal organoid spheroid morphology, mechanistic studies on the functional roles of MMP17-dependent periostin cleavage products, effects of these cleavage fragments on BMP bioactivity and integrin/YAP signaling are not well developed.

In Figure 1 the authors show impressive data related to the gene expression of BMP antagonists Grem1, Grem2 and Chrdn1 in colon smooth muscle tissue compared to crypts. But the organoid co-culture experiment was performed with duodenal organoids and supernatants for colon muscle-SN explants in combination with ENR components. While supernatants from muscle-SN explants display bioactivity that can replace the exogenous BMP antagonist Noggin and supports small intestinal organoid cultures, it would be important to show that colon muscle-SN explants support colon organoids? Or that matched tissues are used for this bioactivity experiments.

In Figure 2 the cell type specific expression of MMP17 based on LacZ and b-gal antibody indicates that it is quite heterogenous and sparse in some regions. FISH data shows that MMP17 overlaps with Grem1 and 2 and Chrdn1 especially in the muscularis mucosae. But what are these MMP17+ cells? SMA appears not to co-localize with B-gal staining – The authors need to provide additional information with cell type specific markers that co-localize with Bgal to identify these MMP17+ cells.

Analysis of MMP17-deficient intestine indicates that MMP17 loss has a greater impact on the colon crypt compartment than the muscle layer with increased nuclear SMAD4 expression and pSmad1/5/9 expression in the crypt cells. What do the different bands represent in the Western blot in Fig 3E. pSMAD1/5/9 appears enriched at the crypt bottom adjacent to the basal lamina but what is its cellular location within these epithelial cells.

Lgr5 and Olfm4 expression is reduced in MMP17-deficient small intestine and this is reflected in decreased organoid formation efficiency from crypts. Did MMP17 loss alter crypt cell proliferation index and distribution of proliferating cells within the crypts? Were there any alterations in cell fate specification within the duodenum or colon? The figure presentation is a little confusing. In Fig 4A, is the quantification on the right for analysis of colon alone? Fig 4C represents colon organoid generation but it was unclear what is represented in Fig 4D. Is this colon organoid efficiency after passaging in WNR and ENR? Not clear why ENR is used but presumably this is duodenal organoids. If so, this should be included in supplemental Fig S4. This significance of figure S4 is also unclear from its title and detail in Figure legend..

For the 10Gy whole body radiation model, the authors comment that "As custom in this model, WT animals regained crypt-villus structures 3 days after irradiation". This is not what be expected for 10Gy whole body radiation. Rather the expectation is to observe regeneration and measure microcolony formation between day 3 and 4 and there should not be not complete renewal of the crypt-villus axis. Clearly the MMP17-deficient appears to be more injured but this analysis is under-developed.

The explanation for the increased tumor initiation in the APCMin tumor model within MMP17-deficient is unclear. With decreased BMP antagonist expression correlated with reduced ISC and OLFM4+ populations as shown in Fig 4, it might be expected that tumor initiation would be expected to decrease. No mechanistic insights are provided for why tumor initiation is increased.

GSEA data for duodenal organoids grown with supernatant from colon muscle-SN explants correlates with a fetal programming signature but the actual data sets used need to be stated and referenced. For example, what YAP gene signature is used? From the images in Fig 7C it is not possible to observe whether there are differences in YAP cellular localization? What are the organoid cultures conditions used for this analysis? Previous data indicated that organoids in muscle-SN explant cultures grew more vigorously and were much larger spheroid structures but images suggest the structures are on similar sizes. More mechanistic studies to understand whether YAP signaling is directly or indirectly involved in spheroid growth and fetal programming. For example, the use of YAP inhibitors and a comparison with organoids cultured with high Wnt activity would be helpful to further dissect out the actions on YAP.

The periostin studies are not well developed. It is surprising the periostin was not detected in gene expression profiling at the beginning of the studies and that other BMP antagonists were not detected by mass spec studies later. While MMP17 can cleave recombinant periostin in vitro no controls are provided for these cleavage studies. The in vivo studies show cleavage fragments that are distinct from in vitro studies – are these relevant and what are the biologically active forms of periostin and do these require MMP17 to be generated. The availability and analysis of periostin-deficient mice would aid these studies.

Addition of recombinant human periostin clearly alters duodenal organoid growth and induces a spheroid morphology. But data provided for Ki67 and YAP staining is poor and low magnification. If periostin binds integrins (which ones?) to activate YAP signaling then it would be expected that the effect would be time dependent and likely early time points for its response. In addition, if periostin binds to BMPs then one might expect that BMP signaling could be manipulated to understand whether this is implicated. Lastly, the dramatic changes in organoid architecture and size/morphology after periostin treatment within matrigel will likely impact mechanosensing that may impact YAP signaling independent of other components. The mechanistic complexity of this response and whether it directs alters or requires YAP signaling is not addressed.

The reference mentioned in the text (Chivukula et al 2014) is not listed in the reference section.

REVIEWER COMMENTS

Reviewer #1 (Remarks to the Author):

Epithelial-mesenchymal interactions are important in the development of the intestinal mucosa, and continue to be essential in post-natal life. Mesenchymal cells such as myofibroblasts and teleocytes (the “pericryptal stroma”) are the sources of growth-regulatory factors for intestinal stem cells, such as the wnts that maintain the critical balance of wnt:BMP signaling. Here, the authors look at “smooth muscle” as an under-studied potential participant in this interaction.

They conclude that smooth muscle express BMP antagonists and that culture supernatants can replace the BMP antagonist NOGGIN in organoid development in vitro.

They select MT4-MMP, known to be expressed in smooth muscle, for further study using a KO mouse. (global; not inducible). While MT4 absence did not detectably affect smooth muscle, there was increased SMAD4 signaling in the KO crypts, interpreted as an absence of BMP antagonists (such as NOGGIN) that arise from mesenchymal cells.

Since this was interpreted to suggest a role for smooth muscle-derived MT4 (?), they used the DSS model of colitis in control vs KO mice, showing a decrease in KI-67 labelling of proliferative epithelial cells in the KO animals, similarly, a radiation-induced injury model showed increased injury in H&E-stained sections of intestine, and there was increased tumour initiation in the APC/min x MT4 KO mouse model.

Returning to the supernatant/organoid model, they found that the supernatant may support organoid growth via YAP. Study of potential targets of MT4 identified periostin, and showed its decreased cleavage in the KO mice. Intact periostin was found to be a growth factor for organoids, but the mechanism was not identified, nor the effects of cleavage products. It is unclear if or why periostin needs to be affected by MT4, an effect lost in the KO mice.

We thank the reviewer for their comments. Indeed, we have found that full-length Periostin mildly induces organoid growth and nuclear YAP in organoids (now Fig. 8g-j, previously Fig. 7h-j). In addition, we refer to the available literature that found that Periostin acts on Integrin-alphaV (Itgav) (Bao et al, 2004; PMID: 15093540, Khurana et al, 2016; PMID: 27905395). Furthermore, we now provide information about the exact cleavage sites on Periostin molecule upon MMP17 exposure using mass spectrometry and on how MMP17-POSTN interact using in silico modelling (new Fig. 8 e, f and Fig. S 8 b, c).

Regarding the different effects of cleaved Periostin, we have tried to determine the difference between full-length (fl) and ‘cleaved’ Periostin in the decellularized intestinal ECM scaffolds (iECM) experimental set up (see more below). We find a distinct improvement of the ‘cleaved Periostin’ compared to ‘fl Periostin’ (see Figure below). However, we used the recombinant enzymatic domain of MMP17 (rMMP17) as a control, and this in itself was able to rescue the KO iECM to allow for crypt formation (new Fig. 7e). This is an important finding, as it shows that MMP17’s enzymatic activity is sufficient to rescue de-cellularized KO tissue in its ability to allow for crypt formation. However, it also means that we cannot attribute higher activity to cleaved Periostin compared to full-length Periostin in the iECM experiments shown here (because rMMP17 is present for enzymatic cleavage).

analyzed in two independent experiments.

While interesting, there are a number of major problems in the presentation and design of this work. The lack of consideration of the known roles of mesenchymal cells in interaction with epithelial stem cells is a major problem. Their contribution to the current study makes it difficult to interpret the present outcomes in either control or MT4-KO mice.

The techniques used are powerful but the cell biology and in vivo approaches have significant gaps, where there is no definitive outcome. This prevents the synthesis a clear picture of the multiple findings, each with interesting, but loosely defined significance. Thus, it is hard to see the proposed role(s) of a smooth muscle-derived enzyme (MT4/MMP17) along with expression of a diffusible factor (yet undefined) that is active in vitro.

The definition of “smooth muscle” is unclear throughout – it is probably an error to equate equally the 3 layers of longitudinal, circular and muscularis mucosa, especially. Technically, the method for obtaining smooth muscle (scraping) is known to remove villus tips, but to leave the lowest levels of the epithelium intact, thus allowing the myofibroblasts to continue to influence outcomes.

*We apologize for the confusion. Our smooth muscle isolation technique does not leave the lowest levels of epithelium intact. We completely removed the epithelium under a dissection scope and a clean muscle sample containing the 3 muscle layers was used in the experiments. Indeed, we confirmed in each extraction that there is no epithelium present by directly checking on the microscope and/or by staining with specific markers such as E-cadherin. We now provide this as new data in **new Fig. S1a**. Moreover, we confirm that our samples only contain smooth muscle by desmin/SMA double staining. Smooth muscle cells can be distinguished from myofibroblasts by Desmin staining; smooth muscle cells are Desmin+, whereas myofibroblasts are Desmin- (Pinchuk et al 2010, PMID: 20690004) (**New Fig. S1a**). Our work is further supported by the finding that our isolated tissue can replace Noggin (BMP antagonists presumably coming from smooth muscle cells) but not R-spondin (normally coming from crypt associated myofibroblasts/telocytes).*

I did not see that the key expt of using supernatant from KO-derived smooth muscle to see if this tissue still expressed NOGGIN-like activity, and thus contributed to organoid development. Does the +/- or -/- affect SMAD4 signaling in the organoids?

*We appreciate this comment by the reviewer. We decided to focus in on the repair process where we found that KO epithelium was unable to repair. Indeed, we now also find a lack of transition into a YAP-high reparative state in KO mice in vivo (see **new Fig. 5i**). Furthermore, the induction of a reparative state was also the strongest effect of WT MSN in vitro (i.e. the induction of large ‘reparative’ spheroids in ENR conditions). Therefore, we took the suggestion by the reviewer and tested WT and KO MSN in the sophisticated iECM model of epithelial regenerative crypt formation. Here, we initially found that iECM derived from KO mice was unable to allow for de novo crypt formation, an important aspect of gut epithelial repair (**Fig. 4e in initial submission, now Fig. 4e-g**). Strikingly, we were able to rescue KO iECM by supplementing MSN from WT but not KO mice (**new Fig. 7e**). This indeed indicates that smooth muscle tissue from KO mice is unable to secrete (or cleave) the relevant factor(s). Furthermore, as we mentioned above, supplementation of KO iECM with recombinant enzymatic domain of MMP17 (rMMP17) also rescued the KO iECM (**new Fig. 7e**), providing further proof of principle that enzymatic cleavage by smooth muscle derived MMP17 is an important factor in epithelial repair.*

Can the various animal models be studied with better resolution so that a mechanism is discovered? These are otherwise just observational.

We now provide additional analysis at the time point reparative processes occur (Day 7 after DSS). At this stage epithelial cells are reprogrammed into a high YAP state that is necessary for repair (for example Yui et al, 2018, PMID: 29249464). We indeed find these high-YAP regenerating cells in WT tissue, but not KO tissue (new Fig. 5i). With this mechanistic insight, our story is also better linked to the in vitro results.

Practically, how does MT4 reach the stem cell niche? Was MT4 measured in the control case? Do the supernatants suppress SMAD4/BMP signaling in vitro? What mediates the smooth muscle-derived supportive effect on organoids?

We apologize for not clarifying this properly in the manuscript. MMP17 is a membrane bound MMP, and so itself will not reach the stem cell niche. We now have added 'membrane-bound' to the abstract (line 20). What we propose here is that smooth-muscle expressed MMP17 cleaves proteins that can reach the stem cell niche, such as periostin, which is one of the factors that can help support organoid growth. However, we think there are many factors derived from smooth muscle, also based on our RNA seq and MS experiments. We hope to identify, characterizing, and test these in full in a follow up study as a next chapter in this story.

The decellularized intestinal expts were not well enough defined to be useful.

We now describe this section in more detail (in main text and methods) and provide convincing IF imaging supporting the findings (New Fig. 4e-g and 7e). We think these, and the expanded experiments, form an important part of the manuscript. They specifically show that ECM of the KO is less receptive for de novo crypt formation, and this can be rescued by recombinant MMP17 and WT muscle-SN but not KO muscle-SN (New Fig. 4e-g and 7e). We have now expanded the methods, and the associated manuscript is deposited on bioRxiv that we cite accordingly.

Throughout, there is a tendency to overstate the significance of the findings, which is not supported by the data – eg, line 257, that MMP17 is required for epithelial reparative responses upon injury. The title might need some thought for improved impact –what is an “intestinal ECM niche” ? and it is not clear that it is “define(d)...” by MMP17.

We have tuned down this sentence (now at line 264: 'Together, these data suggest that muscle-specific MMP17 plays a role in intestinal repair processes after damage. '), and, we changed the title for improved impact and focusing on the main findings of the manuscript.

Reviewer #2 (Remarks to the Author):

General Comments

My review for the manuscript NCOMMS-20-27757 “Smooth muscle-specific MMP17 (MT4-MMP) defines the intestinal ECM niche” is focused on the mass spectrometry/proteomics contribution to the manuscript. The mass spectrometry data presented is adequate but more details in the method needs to be included and I have noted some discrepancies.

Specific Comments

1. The Methods and Materials need to be more specific and or edited.
 - a. Was the protein concentration measured after protein was extracted from the muscle for normalization? i.e. by Bradford or other type of protein concentration techniques?

We have now expanded and clarified the methods section for the MS. Normalization was done by weighing the muscle tissue, and we have performed Coomassie staining of protein gels, but have not determined exact protein concentrations.

b. On line 236 you describe a column that does not exist. I believe it should be a 50 mm length column, not 50 cm. And I don't believe this column comes in a 75 μ m ID (nano). Also you indicate an analytical column was used, please confirm – most proteomics is nano-LC. What was the flow rate of the LC? How much sample was injected?

The column dimensions as described are correct. We pack all our pre- and analytical columns in house and these dimensions are quite typical for high-end proteomics experiments.

Indeed, we used a combination of a trap and an analytical column, with 100 and 75 μ m ID, respectively.

Peptides were trapped for 5 min at 5 μ l/min in 100% solvent A (0.1% formic acid in water). separation was performed at a column flow rate of ~300 nl/min (split flow from 0.2 ml/min) and 5 μ l of the sample was injected. This information is now added to the method section.

c. There is no description on how the data was analyzed, what bioinformatic platform was used and what the search parameters were. How was a peak list generated? What database was searched? What was the FDR? What was the criteria to confidently assign a protein ID. This is a major omission.

We appreciate this comment by the reviewer and agree this was a clear oversight from our side. The following information has been added to the method section:

MS data analysis

All raw MS files were searched using MaxQuant software (version 1.6.10.43). MS/MS spectra were searched by Andromeda against a reviewed Uniprot Mus Musculus (17,068 entries, 2020) using the following parameters: trypsin digestion; maximum of three missed cleavages; cysteine carbamidomethylation as fixed modification; oxidized methionine and protein N-terminal acetylation as variable modifications. Mass tolerance was set to 20 ppm for MS1 and 0.5 Th for MS2. The protein and PSM False Discovery Rate (FDR) were set to 1%.

2. Regarding Supplementary Table 1. What are columns S1, S2 and S3? I am assuming these are biological replicates of the smooth muscle? Or technical replicates? For the column "Unique peptides", is that a combination from the 3 apparent replicates?

Indeed, columns are from 3 independent biological replicates (3 different mice), now clarified in the table (Supplementary Table 2). We removed the unique peptides column from this table for clarity.

3. On line 349 of the manuscript the authors indicate the list of proteins was curated to only display proteins known to be growth factors. I wanted to interrogate the data so I searched the Supplemental Table 1 for the proteins listed in Figure 7d and I only find Manf and Postn. I could not find any of the other listed proteins in Supplemental Table 1. Where did the Proteins listed in 7d come from?

We apologize for the confusion, we used gene names in the manuscript figure, and the protein names in the supplementary file. We have now added the gene name, protein ID number, and protein name to the Figure 7d (new Fig. 8A)

4. I note that the Proteomic data has not been uploaded to a data depository.

The mass spectrometry proteomics data have been deposited to the ProteomeXchange Consortium via the PRIDE partner repository with the dataset identifier PXD020561 (MSN supernatant) PXD025770 (POSTN cleavage). This information is now also added to the manuscript.

Reviewer #3 (Remarks to the Author):

The manuscript addresses the contribution of MMP17+ cells within the intestinal muscle layers in regulating the intestinal stem cell niche during normal intestinal homeostasis and several different injury models in vivo and within organoid models. The authors provide several different lines of investigation that are initially focused on the role of BMP antagonists. Some of the MMP17-deficient data is quite striking in relation to changes to responses such as in repair after acute DSS injury. However other areas of the manuscript are less well developed, for example the radiation injury and APCMin tumor models. Rather than investigate these models in more detail, the authors switch to studying of periostin which is detected in muscle-SN by mass spec and its activity. However, beyond showing recombinant periostin induces altered duodenal organoid spheroid morphology, mechanistic studies on the functional roles of MMP17-dependent periostin cleavage products, effects of these cleavage fragments on BMP bioactivity and integrin/YAP signaling are not well developed.

In Figure 1 the authors show impressive data related to the gene expression of BMP antagonists Grem1, Grem2 and Chrdn1 in colon smooth muscle tissue compared to crypts. But the organoid co-culture experiment was performed with duodenal organoids and supernatants for colon muscle-SN explants in combination with ENR components. While supernatants from muscle-SN explants display bioactivity that can replace the exogenous BMP antagonist Noggin and supports small intestinal organoid cultures, it would be important to show that colon muscle-SN explants support colon organoids? Or that matched tissues are used for this bioactivity experiments.

We performed the suggested experiment using matched small intestinal smooth muscle supernatant with small intestinal derived organoids (to test if SI muscle explants would have equal 'activity'), and, indeed we find that also matched tissue has similar bioactivity (i.e. smooth muscle from either small intestine or colon secretes molecules affecting organoid size and change from local Ki67+ crypts to Ki67 throughout) (new Figure S2)

In Figure 2 the cell type specific expression of MMP17 based on LacZ and b-gal antibody indicates that it is quite heterogenous and sparse in some regions. FISH data shows that MMP17 overlaps with Grem1 and 2 and Chrdn1 especially in the muscularis mucosae. But what are these MMP17+ cells?

SMA appears not to co-localize with B-gal staining – The authors need to provide additional information with cell type specific markers that co-localize with Bgal to identify these MMP17+ cells.

We apologize for not clarifying this properly, Bgal only marks nuclei due to the nuclear localization sequence in the genetic construct (Rikimaru et al

PMID: 17825051), whereas SMA is not nuclear. Please see enlarged image that shows the SMA surrounding a Bgal+ cell (arrow as example). Thus, Bgal+ cells are all SMA-positive, and all these SMA-positive cells in the smooth muscle are Desmin-positive and not CD31 positive (see new Figs. S1A and S3b). We reasoned that perhaps this situation changes after DSS (where there are many infiltrating immune cells). Therefore, in addition to the naïve situation, we now also provide new data showing that MMP17 is not expressed in infiltrating immune cells (CD45+) that accumulate during DSS (new Fig. S6f).

Analysis of MMP17-deficient intestine indicates that MMP17 loss has a greater impact on the colon crypt compartment than the muscle layer with increased nuclear SMAD4 expression and pSmad1/5/9 expression in the crypt cells. What do the different bands represent in the Western blot in Fig 3E.

We note the Smad4-sized band, the other bands are, for as far as we know, aspecific bands as also seen in the product sheet of this antibody: (<https://www.cellsignal.com/products/primary-antibodies/smad4-d3r4n-xp-rabbit-mab/46535>)

pSMAD1/5/9 appears enriched at the crypt bottom adjacent to the basal lamina but what is its cellular location within these epithelial cells.

We have tried to optimize the staining protocol (antigen retrieval, blocking / staining buffers, etc), however, we find that at the bottom it is always throughout the cell (i.e. not specifically nuclear nor is it excluded from the nucleus).

Lgr5 and Olfm4 expression is reduced in MMP17-deficient small intestine and this is reflected in decreased organoid formation efficiency from crypts. Did MMP17 loss alter crypt cell proliferation index and distribution of proliferating cells within the crypts? Were there any alterations in cell fate specification within the duodenum or colon?

We did not observe differences in Ki67+ proliferating cells under naïve conditions (see Figure below), this data is a bit hidden but is shown in the DSS figure (proliferation in naïve animals Fig. 5h). In addition, we now performed Muc2, and Lysozyme staining to assess cell fate specification in naïve animals and did not observe any differences during homeostasis (new Figure S4b).

RevFig2: Equal crypts proliferation in WT and KO intestines. Representative maximal projection of confocal images of distal colon from WT and KO mice stained for the proliferative marker Ki67 (Green), SMA (Red) and DAPI (Blue). Scale 500 μ m; 100 μ m in magnified views. Graph shows % of proliferative cells (Ki67 positive) in mucosa, normalized to total mucosal cell number. n= 7-8 mice per genotype. Numerical data are means \pm SD and were analyzed by Tukey's Multiple Comparison test.

The figure presentation is a little confusing. In Fig 4A, is the quantification on the right for analysis of colon alone? Fig 4C represents colon organoid generation but it was unclear what is represented in Fig 4D. Is this colon organoid efficiency after passaging in WNR and ENR? Not clear why ENR is used but presumably this is duodenal organoids. If so, this should be included in supplemental Fig S4. This significance of figure S4 is also unclear from its title and detail in Figure legend.

Apologies for the confusion, we have now changed all these aspects: Fig. 4A is both colon and small intestine, now indicated in the graph axis (Fig. 4a). Fig. 4d is also colon organoids, however, this to show that after passaging, or growth in factor rich media (WNR), organoids KO efficiency is restored. This is now clarified in Fig. 4 legend. Thus, from this and other experiments we conclude that the in vivo niche is different in KO mice, and that it is not something intrinsic in KO epithelium.

For the 10Gy whole body radiation model, the authors comment that “As custom in this model, WT animals regained crypt-villus structures 3 days after irradiation”. This is not what be expected for 10Gy whole body radiation. Rather the expectation is to observe regeneration and measure microcolony formation between day 3 and 4 and there should not be not complete renewal of the crypt-villus axis. Clearly the MMP17-deficient appears to be more injured but this analysis is under-developed.

The irradiation model likely differs between different labs, locations, and instruments used. We have indeed also found that at a different institute the mice are in full reparative state at day 3. Nevertheless, we have determined the injury level at day 1 (equal cleaved-Caspase-3 / apoptosis, Fig. S6g), and we performed additional mouse experiments and assessed repair 6 days after irradiation. We find that at day 6, the KO still has higher pathology and less repair has occurred. This is now included and combined with the day 3 data in new Fig. 5j and 5k.

The explanation for the increased tumor initiation in the APCMin tumor model within MMP17-deficient is unclear. With decreased BMP antagonist expression correlated with reduced ISC and OLFM4+ populations as shown in Fig 4, it might be expected that tumor initiation would be expected to decrease. No mechanistic insights are provided for why tumor initiation is increased.

In a similar line as the reviewer, we indeed had anticipated lower tumor burdens. We could speculate that the lack of a reparative response in KO mice (as observed in DSS and irradiation models) may account for the increased tumor incidence. However, although we don't have a clear answer of the underlying mechanisms of increased epithelial tumor burden in KO mice, we feel it's an important data set supporting our general message that smooth muscle MMP17 affects intestinal epithelial biology. We agree that further studies of the role of MMP17 in tumor initiation in this context is a very appealing topic that we would like to address in follow up work.

GSEA data for duodenal organoids grown with supernatant from colon muscle-SN explants correlates with a fetal programming signature but the actual data sets used need to be stated and referenced. For example, what YAP gene signature is used?

We now reference the specific papers and have listed the specific genes from the different gene sets used in a new supplementary table 1. Specifically, the YAP signature comes from Gregorieff et al, 2015 (PMID: 26503053) and was made by combining transcriptomics using 'UP' in Yap overexpression and 'DOWN' in Yap-KO organoids.

From the images in Fig 7C it is not possible to observe whether there are differences in YAP cellular localization. What are the organoid cultures conditions used for this analysis? Previous data

indicated that organoids in muscle-SN explant cultures grew more vigorously and were much larger spheroid structures but images suggest the structures are on similar sizes.

We now replaced the images and clarified in the figure legend that we used ENR vs muscle-SN. Indeed, muscle-SN explant organoid co-cultures consistently grow 2-3 fold larger and in spheroid instead of organoid morphology. However, sometimes during fixation etc. they collapse and thus appear not as large during imaging. To obtain better images, we repeated these experiments and now have clearer images showing nuclear YAP in the Muscle-SN treated organoids (Fig. 7c).

More mechanistic studies to understand whether YAP signaling is directly or indirectly involved in spheroid growth and fetal programming. For example, the use of YAP inhibitors and a comparison with organoids cultured with high Wnt activity would be helpful to further dissect out the actions on YAP.

We appreciate these suggestions by the reviewer, and agree that these are useful side-by-side experiments. We thus directly compared CHIR (high Wnt due to GSK3 inhibition) and muscle-SN treated organoids, and find that muscle-SN organoids grow larger (new Fig. 7d). This is supported by using a CHIR gene set (genes UP after CHIR treatment compared to ENR) that has a negative correlation with muscle-SN treated organoids (Fig. 7b). In addition, we tested whether the muscle-SN growth was dependent on YAP by using the YAP inhibitor Verteporfin (to disturb YAP-TEAD interaction). Indeed, we find that Verteporfin completely inhibits muscle-SN-induced growth (new Fig. 7d). Thus, we characterize that these muscle-SN-treated spheroids are quite different from high-WNT spheroids, and that they depend on YAP-co-activated transcription.

The periostin studies are not well developed. It is surprising the periostin was not detected in gene expression profiling at the beginning of the studies and that other BMP antagonists were not detected by mass spec studies later.

Periostin was not picked up by our RNA seq profiling as we selected on growth or growth-related factors, and periostin is known as a 'matricellular' protein. However, it is clearly detected by our RNA seq experiment (300 fold increase, new Fig. 8b). We think the reason BMP antagonists were not found by MS is likely a technical issue, MS is most likely to pick up high abundance proteins.

While MMP17 can cleave recombinant periostin in vitro no controls are provided for these cleavage studies. The in vivo studies show cleavage fragments that are distinct from in vitro studies – are these relevant and what are the biologically active forms of periostin and do these require MMP17 to be generated. The availability and analysis of periostin-deficient mice would aid these studies.

We agree with the reviewer that Periostin-deficient mice would be beneficial to this study, unfortunately, we don't have these mice available at this time, and the development of these studies will require longer time and money. We hope that upon funding this is something we can address in the future. We have however, extended our studies on Periostin as suggested by the reviewer. We provide Osteopontin as a known positive control now for our cleavage studies (See new Fig. S8a). And, we identified several cleavage sites by performing additional mass spectrometry experiments (new Fig. 8e and Fig. S8b) as well as model the interaction in silico, with the identification of the most probable cleavage sites occurring in vivo, as included now in the text and in the new Fig. 8f and Fig. S8c. We further think the difference between mouse and human Periostin and the different isoforms can explain the differences in size between the in vivo (mouse) and in vitro cleavage (human) assays.

Addition of recombinant human periostin clearly alters duodenal organoid growth and induces a

spheroid morphology. But data provided for Ki67 and YAP staining is poor and low magnification. If periostin binds integrins (which ones?) to activate YAP signaling then it would be expected that the effect would be time dependent and likely early time points for its response. In addition, if periostin binds to BMPs then one might expect that BMP signaling could be manipulated to understand whether this is implicated. Lastly, the dramatic changes in organoid architecture and size/morphology after periostin treatment within matrigel will likely impact mechanosensing that may impact YAP signaling independent of other components. The mechanistic complexity of this response and whether it directs alters or requires YAP signaling is not addressed.

*We have now provided new data for the images, with higher magnification for the Ki67 and YAP staining (**new Fig. 8i, j**). We find that periostin modestly (compared to muscle-SN) induces increased organoid size. We provide 2 additional references that show that Periostin acts on the AlphaV integrin (Bao et al, 2004; PMID: 15093540, Khurana et al, 2016; PMID: 27905395). However, as periostin is a matricellular protein, it is difficult to make solid claims from the in vitro work. For example, we found that periostin itself is not able to replace Noggin in vitro (data not shown), however, its described ability to bind BMP would allow for an in vivo effect within the matrix. And, as the reviewer notes, mechanosensing in organoid could be an indirect effect. We have therefore attempted to circumvent this by using the iECM rescue experiments, and indeed see that Full length (fl) Periostin subtly aids in crypt formation, and that cleaved Periostin is more potent – however, and also noted in response to Reviewer 1, we find that the enzymatic domain of MMP17 is sufficient (see figure in response to Rev. 1 and **new Fig. 7e**). Thus, to determine the role and function of Periostin fragments, we would need to perform iECM studies in Periostin-deficient mice, that were supplemented with full length or recombinant fragments of Periostin, which we don't have available at this stage, and thus, we hope that this work will be part of a future chapter of our studies.*

The reference mentioned in the text (Chivukula et al 2014) is not listed in the reference section.

Thank you, we have corrected this.

Reviewers' Comments:

Reviewer #1:

Remarks to the Author:

General

The paper has been revised and is improved. However, some problems persist, whose resolution is needed for understanding of the cellular mechanisms that are proposed. In general, the idea that MMP17 is a smooth muscle-derived, smooth muscle-bound and smooth muscle-specific factor that can influence the geographically remote epithelial crypt is difficult to grasp. A mechanism based on proteolysis of diffusible factor(s) seems required, and if in agreement, wording to this end could be included in the title and abstract, at least.

Major

I Role of smooth muscle.

There is a loose and variable use of "smooth muscle" and "muscularis mucosa" throughout. It is unclear whether the authors view these as synonymous. This should also be clarified. Further, both are distinct from the myofibroblasts in the submucosa that are already known sources of mesenchymal factors that influence epithelial cell growth and maturation. This requires consideration of their separate contributions and potential overlap, and revisions to the text to permit understanding of their roles in these experiments. This was mentioned in the rebuttal, but also belongs in the text. Indeed, I note that the word "myofibroblast" is absent from the paper, and this really should be part of a complete Intro and Discussion.

Examples include Results line 78 "separated smooth muscle from the mucosa". Also, Results line 140-143, muscularis mucosa vs longitudinal and circular smooth muscle; line 162 "enriched in the MM".

Line 162 - "since MMP17 is enriched in the MM, ..."

Despite the inclusion of the MM in the text in several places, it appears that intact smooth muscle consisting only of long. + circ. layers was used for culturing. In Methods, line 549: "mucosal tissue was completely removed" (see also line 78 of Results for a similar statement) - does this mean the MM was excluded from the resulting smooth muscle preparation?

II Smooth muscle in vitro and conditioned medium

The dissection seems to yield "2-4 cm pieces" (line 551), but this requires an area not a length for understanding. How much tissue was used to generate conditioned medium? What volume of medium? As it stands, this expt cannot be reproduced, and the text needs revision for both comprehension and reproducibility.

III iECM

The description of the preparation and nature of the acellular matrix is inadequate. It is said to be a novel technique (line 205) but is not described in the Methods. It is referenced to a pre-publication archive (ref33).

IV MMP17 proteolysis and outcomes

I was unclear as to the relevance of osteopontin (Results, line 391) as an additional MMP17 substrate, to the current results. Clearly, both osteopontin and periostin are degraded by MMP17. The overall cellular mechanism could be clarified for the normal vs inflamed/regenerative states in the adult: it is asserted that MMP17 is required for the reparative state to be fully initiated, but also that periostin does the same thing. However, both osteopontin and periostin are broken down by MMP17. It should be made more clear that this increases or at least does not decrease function. This was mentioned in the rebuttal but is not clear in the text.

Reviewer #2:

Remarks to the Author:

The authors have sufficiently addressed my concerns and I am in favor of publication.

Reviewer #3:

Remarks to the Author:

This is a revised manuscript by Martin-Alonso et al describing the role of MMP17 in regulating intestinal stem cell niche and during intestinal regeneration. The authors have provided significant

additional data and technical information but several lines of investigation are still not integrated well and some conclusions in the discussion appear overstated
Major points.

The authors compare differences in gene expression between intestinal smooth muscle cells and crypt epithelium to demonstrate that BMP antagonists such as Gremlin are abundantly expressed in the smooth muscle layers. While this is true, the authors did not detect Gremlin or other BMP antagonists in the stromal cell compartment of the intestine that is in closer proximity to the epithelium. Indeed, McCarthy et al showed that stromal cells express Gremlin and can readily support organoid growth. This suggests that the RNAscope presented may not be as sensitive as described in the McCarthy publication. The authors in the discussion indicate that the Gremlin-Cre ablation studies from the McCarthy publication support their own work without discussing McCarthy publication in its correct context. The authors state "our work here as well as recent work by others highlight the importance of smooth muscle for providing niche factors important for ISC homeostasis". A more balanced interpretation of the importance of stromal cell compartment and smooth muscle cells is warranted.

There are also some concerns related to the DSS injury model in which increased injury is observed after the withdrawal of DSS in both WT and MMP17 KO mice. If as the authors state the injury is the same at Day 5 DSS then show all the parameters listed in Fig6 within Fig S5. If injury continues in WT after DSS withdrawal as suggested by their data one might presume that increased injury occurs in MMP17 KO as well. If injury is actually more severe in MMP17 KO then it is not surprising that repair is delayed. What is the nadir of injury in the model and how can this be control between WT and MMP17 KO mice?

The role of POSTN, although it is a substrate for MMP17, is unclear and this area of investigation is still not well developed. WT muscle-SN increases organoid area by ~400% whereas POSTN alone only modestly increased organoid area by 25%. The relevance of POSTN to the overall biology of MMP17 activity is clearly overstated. It is unclear whether this line of investigation is still too premature to include in a manuscript focused on MMP17.

Minor points.

For organoid experiments there is still insufficient detail related to how these experiments were performed. For small intestinal organoid experiments, the authors need to state in the results text that they are small intestinal organoids and from what region of they are derived.

For colonoid experiments, the authors should make clear that these are identified as colonoids or colon organoids. In Fig 4c, plate efficiency experiments were performed in ENR + WNT. Fig 4d, it is the same media. But what is "enriched medium WNR" – it is not stated in the methods how this is different from ENR + WNT used in Fig 4c and left panel of 4D. In Fig S5 legend – indicate that small intestinal organoids were grown in ENR media alone.

In Figure 5. Higher magnification images are required to show Ki67 and YAP staining in 5g and 5i, respectively.

Microcolony assay data needs to presented for radiation studies in Fig 5K as a direct measure of regeneration activity.

In Fig 7. Why is YAP staining focused on a region equivalent to the "villus compartment" of the organoid rather than Yap expression in the budding regions which are equivalent to the crypt region? Did treatment with Verteporfin alter CHIR treated organoids as well? What is the purpose of the rMMP17 experiment since MMP17 is membrane-bound with smooth muscle cells and cannot access iECM? Would pretreatment of iECM with MMP17 impact its effect?

REVIEWER COMMENTS

Reviewer #1 (Remarks to the Author):

General

The paper has been revised and is improved. However, some problems persist, whose resolution is needed for understanding of the cellular mechanisms that are proposed. In general, the idea that MMP17 is a smooth muscle-derived, smooth muscle-bound and smooth muscle-specific factor that can influence the geographically remote epithelial crypt is difficult to grasp. A mechanism based on proteolysis of diffusible factor(s) seems required, and if in agreement, wording to this end could be included in the title and abstract, at least.

We thank the reviewer for this comment, and we agree that this ('proteolysis of diffusible factor(s)') is the mechanism we propose. We have added a sentence highlighting the mechanism based on proteolysis of diffusible factor(s) to the abstract of the manuscript.

Major

I Role of smooth muscle.

There is a loose and variable use of "smooth muscle" and "muscularis mucosa" throughout. It is unclear whether the authors view these as synonymous. This should also be clarified. Further, both are distinct from the myofibroblasts in the submucosa that are already known sources of mesenchymal factors that influence epithelial cell growth and maturation. This requires consideration of their separate contributions and potential overlap, and revisions to the text to permit understanding of their roles in these experiments. This was mentioned in the rebuttal, but also belongs in the text. Indeed, I note that the word "myofibroblast" is absent from the paper, and this really should be part of a complete Intro and Discussion.

Examples include Results line 78 "separated smooth muscle from the mucosa". Also, Results line 140-143, muscularis mucosa vs longitudinal and circular smooth muscle; line 162 "enriched in the MM". Line 162 – "since MMP17 is enriched in the MM, ..."

Despite the inclusion of the MM in the text in several places, it appears that intact smooth muscle consisting only of long. + circ. layers was used for culturing. In Methods, line 549: "mucosal tissue was completely removed" (see also line 78 of Results for a similar statement) - does this mean the MM was excluded from the resulting smooth muscle preparation?

*We have added the information from the previous rebuttal about known niche factor sources from mesenchymal cells into the introduction for completeness (line 51 and following, and **FigS1a** figure legend). We apologize for the confusion, now we have stated throughout the text what we refer to specifically. Most of the time this is smooth muscle cells (when we cannot distinguish), but sometimes we specify its source (muscularis propria and/or muscularis mucosae in each case). In addition, we have clarified that the muscularis mucosae is part of the mucosa, and corrected all the sentences mentioned by the reviewer, including the methods part. Finally, we note that the smooth muscle preparation includes the muscularis propria and patches of muscularis mucosae as showed in **FigS1a**.*

II Smooth muscle in vitro and conditioned medium

The dissection seems to yield "2-4 cm pieces" (line 551), but this requires an area not a length for

understanding. How much tissue was used to generate conditioned medium? What volume of medium? As it stands, this expt cannot be reproduced, and the text needs revision for both comprehension and reproducibility.

As suggested by the reviewer, we have corrected the mistake on the methods regarding muscle samples isolation and muscle-SN collection (see methods section on page 19) and carefully reviewed this methods section. We have added now new information on the protocol that we hope will help in the comprehension and reproducibility of the method.

III iECM

The description of the preparation and nature of the acellular matrix is inadequate. It is said to be a novel technique (line 205) but is not described in the Methods. It is referenced to a pre-publication archive (ref33).

*Clarification on this method has been also provided in results and in the methods section (page 29-30), and added an improved schematic on the protocol on **Fig 4e** and an explanation on this figure legend. We have properly cited the article (ref 39). We are sorry about the confusion, since we meant that this iECM regeneration assay is a recently developed technique.*

IV MMP17 proteolysis and outcomes

I was unclear as to the relevance of osteopontin (Results, line 391) as an additional MMP17 substrate, to the current results. Clearly, both osteopontin and periostin are degraded by MMP17. The overall cellular mechanism could be clarified for the normal vs inflamed/regenerative states in the adult: it is asserted that MMP17 is required for the reparative state to be fully initiated, but also that periostin does the same thing. However, both osteopontin and periostin are broken down by MMP17. It should be made more clear that this increases or at least does not decrease function. This was mentioned in the rebuttal but is not clear in the text.

We thank the reviewer for this comment. We added osteopontin cleavage in the revision process as it was requested by the reviewer as a positive control for MMP17 catalytic activity in the POSTN digestion assay.

Furthermore, now we have clarified in the discussion to highlight the activation step of POSTN upon cleavage (Line 335)

Reviewer #2 (Remarks to the Author):

The authors have sufficiently addressed my concerns and I am in favor of publication.

Reviewer #3 (Remarks to the Author):

This is a revised manuscript by Martin-Alonso et al describing the role of MMP17 in regulating intestinal stem cell niche and during intestinal regeneration. The authors have provided significant additional data

and technical information but several lines of investigation are still not integrated well and some conclusions in the discussion appear overstated

Major points.

The authors compare differences in gene expression between intestinal smooth muscle cells and crypt epithelium to demonstrate that BMP antagonists such as Gremlin are abundantly expressed in the smooth muscle layers. While this is true, the authors did not detect Gremlin or other BMP antagonists in the stromal cell compartment of the intestine that is in closer proximity to the epithelium. Indeed, McCarthy et al showed that stromal cells express Gremlin and can readily support organoid growth. This suggests that the RNAscope presented may not be as sensitive as described in the McCarthy publication.

We thank the reviewer for this comment. We believe that the differences regarding Grem1 RNAscope in McCarthy et al publication and our results arise from the different tissue source. Our RNAscope is performed using colonic mouse tissue while in McCarthy et al, they used small intestine. Stromal cell populations, levels and types of factors secreted by these cells could considerably vary between colon and small intestine. Nevertheless, as shown in Fig 2d and highlighted in the following inset of the image, we find some Grem1 expression in cells that are outside of the smooth muscle cells region and closer to the crypts (magenta spots in RNAscope image

are outside of the smooth muscle cells region and closer to the crypts (magenta spots in RNAscope image (yellow arrows)). C means Crypts, MM, muscularis mucosae, and M, muscularis propria.

Moreover, we would like to mention that the intensity in RNAscope expression of BMP antagonists by smooth muscle cells is considerably higher when compared to its expression in the lamina propria (putatively by stromal cells).

The authors in the discussion indicate that the Gremlin-Cre ablation studies from the McCarthy publication support their own work without discussing McCarthy publication in its correct context. The authors state “our work here as well as recent work by others highlight the importance of smooth muscle for providing niche factors important for ISC homeostasis”. A more balanced interpretation of the importance of stromal cell compartment and smooth muscle cells is warranted.

We apologize for the confusion. We have now clarified this aspect in the discussion (line 322 and following), and we have added a line noting that future work is warranted to determine the importance of these different cell types and their ability to provide niche factors.

There are also some concerns related to the DSS injury model in which increased injury is observed after the withdrawal of DSS in both WT and MMP17 KO mice. If as the authors state the injury is the same at Day 5 DSS then show all the parameters listed in Fig6 within Fig S5. If injury continues in WT after DSS withdrawal as suggested by their data one might presume that increased injury occurs in MMP17 KO as

well. If injury is actually more severe in MMP17 KO then it is not surprising that repair is delayed. What is the nadir of injury in the model and how can this be control between WT and MMP17 KO mice?

Day 5 is usually considered the control point in this type of assay since it is when the injury trigger is removed. As previously described, 5 days of DSS exposure generates “weight loss and severe acute colitis, leaving patches of completely de-epithelialized mucosa” that is followed by a regenerative process after DSS removal (Chivukula et al, 2014, PMID: 24855947). This regeneration involves the resolution of inflammation that further could affect the tissue for the 2 next days. At day 7 (2 days after DSS removal), WT regeneration is represented by elongated hyperproliferative crypts, however tissue still shows heavily damaged areas where inflammation is resolving. We believe that in the KO the inability to restore the epithelial barrier, in other words, to regenerate, allows for prolonged inflammation that could (perhaps) cause damage during longer time. So, KO colon would be suffering a combination of impaired regeneration and inflammation that make them show as damaged and inflamed as in the images of Fig 5.

We have now added an injury score graph on Day 5 in new FigS 6c to show all the parameters of Fig 5. However, we think the YAP staining is not necessary at Day 5 since YAP is induced after DSS removal and is induced at Day7 in WTs (not day 5) as previously shown by Cai et al. 2010. PMID: 21041407.

The role of POSTN, although it is a substrate for MMP17, is unclear and this area of investigation is still not well developed. WT muscle-SN increases organoid area by ~400% whereas POSTN alone only modestly increased organoid area by 25%. The relevance of POSTN to the overall biology of MMP17 activity is clearly overstated. It is unclear whether this line of investigation is still too premature to include in a manuscript focused on MMP17.

We thank the reviewer for this comment, and we have toned down the interpretation of the POSTN data in the results and discussion especially relative to the activity we observed by Muscle-SN (discussion line 335 and following). Nevertheless, we think that finding a new substrate for the protease, that is specifically expressed by smooth muscle cells and that is able to affect epithelial behavior (as inducing regenerative YAP-linked activity) is important enough to maintain this part of the data in the manuscript. Moreover, describing a new substrate and putative cleavage sites will help in the investigation of MMP17 roles in other important contexts as cancer, inflammation, or in other organs/tissues, and thus well-suited for the broad audience of Nat Comms.

Minor points.

For organoid experiments there is still insufficient detail related to how these experiments were performed. For small intestinal organoid experiments, the authors need to state in the results text that they are small intestinal organoids and from what region of they are derived.

For colonoid experiments, the authors should make clear that these are identified as colonoids or colon organoids. In Fig 4c, plate efficiency experiments were performed in ENR + WNT. Fig 4d, it is the same media. But what is “enriched medium WNR” – it is not stated in the methods how this is different from ENR + WNT used in Fig 4c and left panel of 4D. In Fig S5 legend – indicate that small intestinal organoids were grown in ENR media alone.

We apologize for the confusion and now have clearly stated in each case if we were using small intestine (SI)-derived organoids or colonoids through the text, methods, legends, and supplementary data. We hope these changes further help to understand the experiments performed.

In Figure 5. Higher magnification images are required to show Ki67 and YAP staining in 5g and 5i, respectively.

We have now added an inset on both panels (Fig 5g and i) showing specific cellular pattern for Ki67 and YAP staining at a higher magnification.

Microcolony assay data needs to be presented for radiation studies in Fig 5K as a direct measure of regeneration activity.

We thank the reviewer for this suggestion, and we have now included a quantification of the number of crypts/tissue length performed on H&E swiss roll pictures of Day3 and Day6 irradiation studies (new Fig 5k).

In Fig 7. Why is YAP staining focused on a region equivalent to the “villus compartment” of the organoid rather than YAP expression in the budding regions which are equivalent to the crypt region?

The purpose of this staining is to show activation of YAP after muscle-SN exposure. YAP is continuously active in organoid crypts whereas it is commonly inactive in the villus compartment of organoids. Muscle-SN-derived spheroids are devoid of proper crypt structures and mainly composed of these areas with nuclear positive YAP. Other spheroid subtypes, such as ‘enterocysts’ do not have these high YAP levels (Serra et al Nature 2019, PMID 31019299)

Did treatment with Verteporfin alter CHIR treated organoids as well?

Verteporfin indeed affects CHIR treated organoids completely blocking its growth. This is similar as previously published by others: organoids with impaired Wnt/BCatenin signaling (ApckO) do not develop when exposed to verteporfin, showing YAP requirement in response to exacerbated Wnt signaling (Yui et al, 2018. PMID: 29249464).

What is the purpose of the rMMP17 experiment since MMP17 is membrane-bound with smooth muscle cells and cannot access iECM?

In this experiment, we added the catalytic domain of MMP17 (rMMP17) in soluble form. In this way, rMMP17 is able to access the ECM matrix of the iECM and affect the ECM in which the epithelial cells are grafting. In such manner, we are able to probe that MMP17 catalytic activity is a regulator of epithelial regeneration. We do agree, however, that in vivo, MMP17 would be in the smooth muscle cells membrane, but still able to access many substrates that will impact into epithelial behavior.

Would pretreatment of iECM with MMP17 impact its effect?

We think it might, and it's a good suggestion – we are planning follow-up work and are trying to acquire funding and hope to provide an answer in the future.

Reviewers' Comments:

Reviewer #1:

Remarks to the Author:

This revised version has responded to the criticisms arising from my previous review, largely by providing missing details that elaborate on the methods used, and by modifying some conclusions. Despite critique (my review 1: "The lack of consideration of the known roles of mesenchymal cells in interaction with epithelial stem cells is a major problem."), it is still difficult to identify a cellular mechanism that uses the separate observations that are made here, (my review 2: "...resolution is needed for understanding of the cellular mechanisms that are proposed).

How, and why, membrane bound MMP17 acts to affect the geographically remote epithelial compartment is still unclear. This is not considered in the light of the known intimate association of myofibroblasts and other mesenchymal cell types (eg, McCarthy, Cell Stem Cell 2020) that have established trophic roles. Indeed, the role of GREM1 in intestinal myofibroblasts (Zhang, Sci Rep 2018 PMID 30120338) and other factors (eg, Rowan, J Pathol 2020; reviewed Mendena&Vermeulen, Nature 2011) are established. The membrane-bound site of MMP17 was also raised in the recent response to review; here again, its local (stromal compartment) actions on substrates of relevance to the epithelial crypt may occur, but this activity and the consequences are not demonstrated.

Some data continues to be only weakly supportive, such as the role of POSTN as a growth factor (Fig 8(g-j)). This shows a "modest increase" at best and indeed, most of the data overlap between control and +POSTN. Here, the evidence for a coordinate increase in KI67 and YAP is qualitative and does not support a major role.

Reviewer #3:

Remarks to the Author:

The authors have attempted to address most of the concerns in the revised manuscript and the manuscript is improved.

REVIEWERS' COMMENTS

Reviewer #1 (Remarks to the Author):

This revised version has responded to the criticisms arising from my previous review, largely by providing missing details that elaborate on the methods used, and by modifying some conclusions. Despite critique (my review 1: "The lack of consideration of the known roles of mesenchymal cells in interaction with epithelial stem cells is a major problem."), it is still difficult to identify a cellular mechanism that uses the separate observations that are made here, (my review 2: "...resolution is needed for understanding of the cellular mechanisms that are proposed).

How, and why, membrane bound MMP17 acts to affect the geographically remote epithelial compartment is still unclear. This is not considered in the light of the known intimate association of myofibroblasts and other mesenchymal cell types (eg, McCarthy, Cell Stem Cell 2020) that have established trophic roles. Indeed, the role of GREM1 in intestinal myofibroblasts (Zhang, Sci Rep 2018 PMID 30120338) and other factors (eg, Rowan, J Pathol 2020; reviewed Mendena&Vermeulen, Nature 2011) are established.

We thank the reviewer for these comments. We propose that the impaired POSTN cleavage causes an alteration in the ECM/cell signaling that would affect the epithelium. We agree that the specificities of this possible interaction (through interaction to specific ECM components for example) would be of extreme interest, but too difficult to address in this manuscript. We are aware of the role of myofibroblasts and mesenchymal cells in regulating the ISCs niche, and we highlight this in the introduction and discussion. However, we believe that the discovery of smooth muscle cells role in epithelial regulation by expression of high levels of these BMP antagonists, defining them as a new source, is of importance to the field and in part a focus of our study.

The membrane-bound site of MMP17 was also raised in the recent response to review; here again, its local (stromal compartment) actions on substrates of relevance to the epithelial crypt may occur, but this activity and the consequences are not demonstrated.

We do agree with the reviewer about the different localization of smooth muscle/crypts, however, we provide enough data along the manuscript that demonstrate the impact of the smooth muscle into ISCs crypts niche. For example, iECM studies shows that the physical/cellular niche its completely different in KO mice and indeed we can rescue the niche by adding only WT muscle SN.

Even though MMP17 is restricted to the smooth muscle cell membrane, the release of different cleavage products to the extracellular media would affect the physical and cellular environment surrounding the crypt. Indeed, it is known that POSTN and perhaps its cleavage-derived products could differently interact to BMP1 resulting in alterations in Fibronectin deposition and therefore changing the surrounding ECM.

Some data continues to be only weakly supportive, such as the role of POSTN as a growth factor (Fig 8(g-j)). This shows a "modest increase" at best and indeed, most of the data overlap between control and +POSTN. Here, the evidence for a coordinate increase in KI67 and YAP is qualitative and does not support a major role.

We thank the reviewer for this comment. We do agree that the effect of POSTN in organoids is modest compared to Muscle SN effect, however, as proved in the MS data, Muscle SN produces many

other growth factors that contribute to the impressive phenotype observed in organoids. So it is not surprising in our opinion that the cumulative effect of Muscle SN produced factors is bigger than POSTN molecule alone. Moreover, activation of nuclear YAP signaling and the changes on organoids shape and KI67 distribution shows that POSTN is contributing to regulate the main changes observed in Muscle SN treated organoids.

As mentioned in previous reviews, cleaved POSTN would presumably have stronger effect regulating organoids shape and size, more than the Full-length POSTN used for the experiment. Future experiments using Lentiviral vectors codifying for FL-POSTN or the different fragments generated after MMP17-cleavage will be of interest to demonstrate the specific effect of POSTN in the epithelium.